



# What caused a record high PM$_{10}$ episode in northern Europe in October 2020?

Christine D. Groot Zwaaftink, Wenche Aas, Sabine Eckhardt, Nikolaos Evangeliou, Paul Hamer, Mona Johnsrud, Arve Kylling, Stephen M. Platt, Kerstin Stebel, Hilde Uggerud, Karl Espen Yttri

NILU - Norwegian Institute for Air Research, P.O. Box 100, N-2027 Kjeller, Norway

*Correspondence to*: Christine Groot Zwaaftink (cgz@nilu.no)

## Abstract

Early October 2020, northern Europe experienced an episode with poor air quality due to high concentrations of particulate matter (PM). At several sites in Norway, maximum recorded values for weekly averaged PM$_{10}$ concentrations from the past 4

to 10 years were exceeded. Daily mean PM$_{10}$ values at Norwegian sites were up to 97 µg m$^{-3}$ and had a median value of 59 µg m$^{-3}$. We analysed this severe pollution episode caused by long-range atmospheric transport based on on-line and off-line surface and remote sensing observations and transport model simulations to understand its causes. Samples from three sites in mainland Norway and the Arctic remote station Zeppelin (Svalbard) showed strong contributions from mineral dust to PM$_{10}$ (23-36% as a minimum and 31-45% as a maximum) and biomass burning (8-16% – 19-21%). Atmospheric transport

simulations indicate that Central Asia was the main source region for mineral dust observed in this episode. The biomass burning fraction can be attributed to forest fires in Ukraine and southern Russia, but we cannot exclude other sources contributing as well. The combined use of remote sensing, high quality measurements and transport modelling proved effective in describing the episode and distinguishing its causes.

## 1 Introduction

During a two-day's episode in early October 2020, atmospheric concentrations of PM$_{10}$ and PM$_{2.5}$, i.e. particulate matter (PM) with an equivalent aerodynamic diameter of ≤10 µm and ≤2.5 µm, were elevated at many stations in northern Europe (Fig. 1), exceeding international and national air quality guidelines (AQG) (WHO, 2006). On 2 and/or 3 October, daily mean EU-limit values for PM$_{10}$ of 50 µg m$^{-3}$ were exceeded at 45 of the European Environment Agency's (EEA) air quality urban and regional background sites (Fig. 1). In Norway, daily averaged PM$_{10}$ values exceeded previous maximum values recorded at several

sites. Visibility was strongly reduced, and questions concerning the source of these pollutants and possible influence on health arose (e.g. Aftenposten, 2020).

PM levels are usually increased in winter and spring compared to summer in urban areas in northern Europe, largely explained by increased emissions from residential wood burning and resuspension of road dust generated by use of studded tires, contributing to the fine and the coarse fraction of PM$_{10}$, respectively (Kukkonen et al., 2020; Laupsa et al., 2009; Yttri et al.,





2009; Yttri et al., 2005). However, in this case the widespread exceedance of PM$_{10}$ values, not only in Norway, pointed to long-range transport (LRT) of PM. Violation of limit values caused by LRT PM in Norway rarely occur. Historically, high levels of LRT PM have been associated with secondary inorganic aerosol species, SO$_4^{2-}$, NO$_3^-$ and NH$_4^+$, formed by atmospheric oxidation of European SO$_2$, NH$_3$ and NO$_x$ emissions (e.g. Tarrason et al., 2019; Yttri et al., 2021). A fast-track analysis showed that the source of the increased concentrations could have been biomass burning in Eastern Europe. However,

the fact that concentrations of coarse fraction PM$_{10}$ were also enhanced suggested that sources such as dust, ash and sea salt aerosol could also be important. Out of these, mineral dust (from agricultural or natural sources) is the most likely based on the region with enhanced PM levels (Fig. 1).

Events of long-range transport of mineral dust to Scandinavia are rare. Mineral dust outbreaks from the Sahara can reach the surface in Scandinavia (e.g. Ansmann et al., 2003), although the dust plumes likely reside at higher altitudes. An aerosol

transport event with combined contributions from biomass burning from the Iberian Peninsula and dust emissions from North Africa, affecting several regions in Europe, was described by Akritidis et al. (2020). Also, transport events of mineral dust from Kazakhstan have been shown to affect Scandinavia (e.g. Hongisto and Sofiev, 2004). Besides affecting surface concentrations of PM and air quality, airborne dust can strongly influence the radiation budget of the atmosphere, both directly and indirectly (e.g. Kylling et al., 2018; Myhre et al., 2013).

LRT events of aerosol particles from biomass burning have been observed frequently in Scandinavia (e.g., Yttri et al., 2007b; Saarikoski et al., 2007) including the European Arctic (Stohl et al., 2007), and can be related to wildfires and agricultural waste burning in the Baltic countries, western Russia, Belarus, and Ukraine. Fires release various particulate and gaseous substances such as organic and black carbon (OC and BC), CO$_2$, CH$_4$, and polyarometaic hydrocarbons (PAHs) (Hao et al., 2016; Hao and Ward, 1993; Shi et al., 2015), some of which have an adverse effect on human health, such as benzo(a)pyrene, a Group 1

carcinogen (IARC, 2021).

To quantify biomass burning contributions to PM we analyse filter samples for carbonaceous aerosol as well as levoglucosan (a biomass burning tracer). Long-range transport modelling of the biomass burning aerosol in this study is based on BC, which is formed by the incomplete combustion of e.g., fossil fuels, biofuels, and biomass (Bond et al., 2013). BC not only affects human health (Lelieveld et al., 2015), but also climate (Myhre et al., 2013).

Here, we show how LRT of mineral dust from Central Asia and biomass burning aerosol, mainly from Ukraine, caused elevated PM levels in Norway. Our results are based on a combination of in-situ observations, satellite images and model simulations of long-range atmospheric transport.

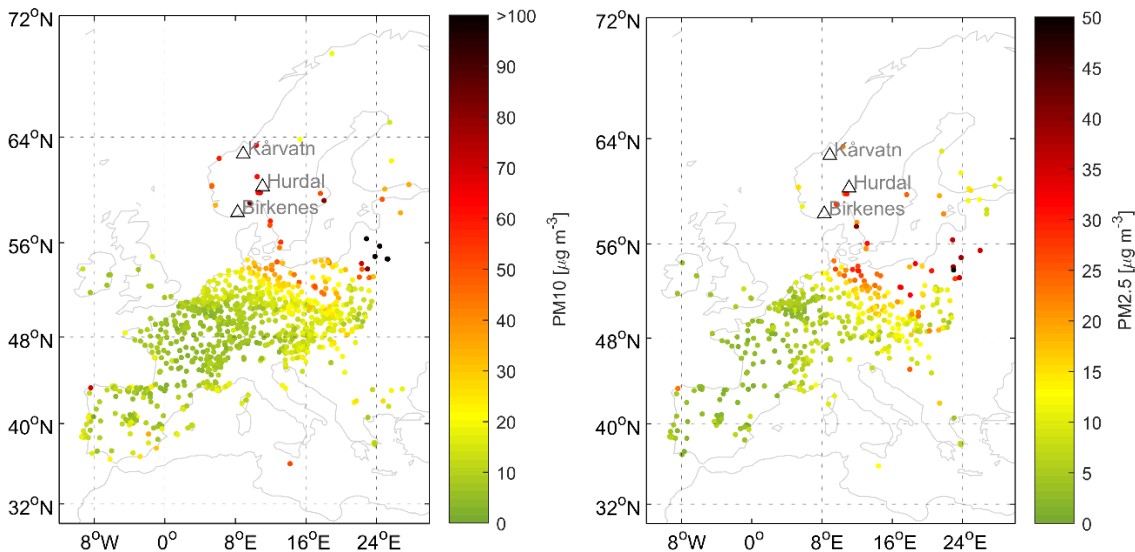

**Figure 1 Mean PM$_{10}$ and PM$_{2.5}$ values on 2 and 3 October 2020 (48-hour mean) at all EEA air quality background sites (includes urban as well as rural, but sites close to roads are excluded). Triangles indicate three Norwegian rural background sites with extensive PM speciation (section 2.1.1). For comparison: The WHO air quality guidelines for maximum daily averaged PM$_{10}$ and PM$_{2.5}$ are 50 µg·m$^{-3}$ and 25 µg·m$^{-3}$, respectively, the EU limit value for PM$_{10}$ is 50 µg·m$^{-3}$.**

## 2. Data and methods

### 2.1 Surface observations

### 2.1.1 Sites and instrumentation

There are 48 (urban categorized) sites in Norwegian cities and towns measuring both PM$_{10}$ and PM$_{2.5}$ mass concentration as part of their air quality programme Hak (2015). Rural background stations include the Birkenes Observatory (58°23'N, 8°15'E, 219 m above sea level, asl), Hurdal (60°22 N, 11° 4'E, 300 m asl) and Kårvatn (62°47'N, 8°53'E, 210 m asl). These are all EMEP (European Monitoring and Evaluation Programme) sites located in southern Norway (Fig. 1). The Birkenes Observatory is situated close to the Skagerrak coast (~20 km), whereas Hurdal and Kårvatn are further inland. Forests dominate the land use at all three sites, the remainder being meadows, low intensity agricultural areas and freshwater lakes. All three sites are thus considered rural background sites for air quality measurements. The Zeppelin Observatory (78°54' N 11°52' E, 472 m asl) is located on the Zeppelin Mountain 2 km south of the Ny-Ålesund settlement at the west coast of Svalbard (Norway) and is considered an Arctic remote/ global background site. PM mass concentration (PM$_{10}$ and PM$_{2.5}$), organic and elemental carbon (OC/EC) (in PM$_{10}$ and PM$_{2.5}$), organic tracers (levoglucosan, mannosan, galactosan, arabitol, mannitol, glucose, trehalose, 2-methylerythritol, and 2-methylthreitol) (in PM$_{10}$), and crustal elements (Al, Fe, Mn, and Ti) (in PM$_{10}$) were obtained from low-volume samplers (flow rate of 38 L min$^{-1}$) with a sampling time of one week at the



rural background sites. Prefired (850 °C; 3 h) quartz fiber filters (Whatman QM-A; 47 mm in diameter) were conditioned (20±1 °C; 50±5% RH (relative humidity); 48 h) before and after exposure. Filters were stored at 4 °C after

weighing and before OC/EC analysis and at -18 °C prior to organic tracer analysis. Please find further details on the analysis methods in appendix A1.

At the Zeppelin Observatory, OC/EC, organic tracers and crustal elements were obtained from a high-volume sampler with a $PM_{10}$ inlet (flow rate of 667 L min$^{-1}$) and with a sampling time of one week. Quartz fiber filters (PALLFLEX Tissuequartz 2500QAT-UP; 150 mm in diameter) were prefired (850 °C; 3 h) but not conditioned, and stored at -18 °C.

Major inorganic anions ($SO_4^{2-}$, $NO_3^-$, $Cl^-$) and cations ($Ca^{2+}$, $Mg^{2+}$, $K^+$, $Na^+$, $NH_4^+$) were collected using a NILU stacked filter unit (SFU) collecting aerosol particles on a Teflon filter (2 µm pore, 47 mm Zefluor Teflon, Gelman sciences) with a sampling time of 24 h. The SFU does not have a pre-impactor but has a downward facing inlet that effectively reduces the sampling efficiency for aerosol particles with an equivalent aerodynamic diameter larger than 10 µm.

$PM_{10}$ mass concentration was measured using a TEOM 1400a (Tapered Element Oscillating Monitor)

(Patashnick and Rupprecht, 1991) instrument operating at a one-minute time resolution at the Birkenes Observatory. The aggregated weekly mean $PM_{10}$ concentration obtained by the TEOM instrument is proven equivalent to the weekly time resolved, filter based $PM_{10}$ measurements at Birkenes, which is determined according to the reference method EN 12341 (CEN, 2014).

Absorption coefficients ($B_{Abs}$) were measured using a seven-wavelength (λ=370; 470; 520; 590: 660; 880; 950 nm)

Aethalometer (AE33, Magee Scientific) operating downstream of a $PM_{10}$ inlet and calculated according to Drinovec et al. (2015) at the Birkenes and Zeppelin observatories. $B_{Abs}$ was converted to eBC using co-located EC measurements.

### 2.1.2 Mineral dust mass estimation

We assumed observed levels of Al, Fe, Mn and Ti to be associated with mineral dust exclusively, and that Al was present as

$Al_2O_3$, Fe as $Fe_2O_3$, Mn as MnO and Ti as $TiO_2$ (Alastuey et al., 2016). Not all mineral dust elements were measured such as Si, nor is the mineralogy of the mineral dust collected on the filter samples known, thus, $SiO_2$ was estimated based on an empirical factor (eq.1) (Alastuey et al., 2016). We also made assumptions regarding the speciation of $CO_3^{2-}$, assuming 70% was present as $CaCO_3$, 20% as $MgCO_3$ and 10% as $K_2CO_3$. $Ca^{2+}$, $Mg^{2+}$, and $K^+$ assumed to be associated with $CO_3^{2-}$ were not considered part of the water-soluble fraction determined by ion chromatography (Appendix 1). Excess $Ca^{2+}$, $Mg^{2+}$, and $K^+$ (i.e.,

not part of sea salt aerosol) were assumed to be present as oxides (CaO, MgO, and $K_2O$). We consider this an approximation, as chemical reactions likely take place during atmospheric transport. Both $NO_3^-$ and $SO_4^{2-}$ were enhanced during the episode and could indicate the presence of $Ca(NO_3)_2$ and $CaSO_4$ formed from reactions between $CaCO_3$ and acids such as $HNO_3$ or $H_2SO_4$ (Laskin et al., 2004).

$CO_3^{2-}$ can also be part of wildfire emissions and is a dominant species of ash produced at around 500 °C with $CaCO_3$ being the

most abundant species followed by $MgCO_3$ and $K_2CO_3$ (Bodí et al., 2014). Hence, it is questionable whether $CO_3^{2-}$ should be





apportioned as mineral dust or biomass burning when such emissions are mixed in the atmosphere. At temperatures >580 ˚C, carbonates dissociate to oxides (Bodí et al., 2014), which are partially soluble in water, thus, as for $CO_3^{2-}$, it is not clear if CaO, MgO, and $K_2O$ should be part of mineral dust or biomass burning particles. Consequently, any attempt to reconstruct the mineral dust mass concentration should be considered semi-quantitative. We thus calculate a lower (eq.2) and an

upper (eq.3) estimate of the mineral dust mass concentration including no or all $CO_3^{2-}$ and oxides, respectively.

$[SiO_2] = 2.5 \times [Al_2O_3]$ (eq.1)

$[Mineral\ dust_{mass}]_{lower} = [SiO_2] + [Al_2O_3] + [Fe_2O_3] + [MnO] + [TiO_2]$ (eq.2)


$[Mineral\ dust_{mass}]_{upper} = [SiO_2] + [Al_2O_3] + [Fe_2O_3] + [MnO] + [TiO_2] + [(n)CO_3^{2-}] + [(n)O]$ (eq.3)

$(n = Ca^{2+},\ Mg^{2+},\ 2Na^+,\ 2K^+)$

**2.1.3 Biomass burning mass estimation**

Levoglucosan is formed from the thermal degradation of cellulose and is a proven tracer of biomass burning (BB) emissions (Locker, 1988; Simoneit et al., 1999) as demonstrated in numerous papers (e.g., Zdrahal et al., 2002; Puxbaum et al., 2007; Szidat et al., 2009; Yttri et al., 2019). Emission ratios of levoglucosan from wildfires are likely to vary widely reflecting combustion conditions and vegetation, and source region. Here we apply emission ratios for total

carbon and organic carbon (Yttri et al., 2014) to calculate $OC_{BB}$ and $EC_{BB}$, which are derived from ambient sampling of wildfires emissions in Eastern Europe, including Ukraine (Saarikoski et al., 2007). Calculated concentrations of $OC_{BB}$ and $EC_{BB}$ should be considered semiquantitative given the uncertainty of the emission ratios and the potential atmospheric depletion of levoglucosan.

Although BB aerosol from wildfires is dominated by carbonaceous aerosol, carbonates and oxides can result from wildfires,

as well as being a part of mineral dust. Hence, we calculated a lower estimate of the biomass burning aerosol accounting for the carbonaceous fraction ($TC_{BB}=OC_{BB}+EC_{BB}$) (eq.4-7) and an upper estimate accounting for carbonates and oxides, in addition to the carbonaceous fraction (eq.8).

$[TC_{BB}] = [Levoglucosan] \times (TC/levoglucosan)_{BB}$    (eq.4)

$[OC_{BB}] = [TC_{BB}] \times (OC/TC)_{BB}$    (eq.5)

$[EC_{BB}] = [TC_{BB}] - [OC_{BB}]$    (eq.6)

$[BB_{mass}]_{Lower} = [OC_{BB} \times 2.2] + [EC_{BB} \times 1.1]$    (eq.7)





$[BB_{mass}]_{Upper} = [OC_{BB} \times 2.2] + [EC_{BB} \times 1.1] + [(n)CO_3^{2-}] + [(n)O]$   $(n = Ca^{2+}, Mg^{2+}, 2Na^+, 2K^+)$ (eq. 8)


where $BB_{mass}$ is the mass concentration (µg m$^{-3}$) of the biomass burning aerosol.

For aerosol particles dominated by biomass burning, a factor of 2.2-2.6 to convert $OC_{BB}$ (µg C m$^{-3}$) to $OM_{BB}$ (organic matter; µg m$^{-3}$) is recommended, whereas 1.9-2.2 is suggested for aged aerosol particles (Turpin and Lim, 2001). Here we used a factor of 2.2 both for $OC_{BB}$ and OC, as OC at rural background and remote sites largely are long-range transported and thus aged. Similarly, a factor of 1.1 was used for both $EC_{BB}$ and EC (Kiss et al., 2002).


We estimated levels of primary biological aerosol particles (PBAP), biogenic secondary organic aerosol (BSOA), and fossil fuel sources (FF), as these are complementary to BB and yield a better constraint on the biomass burning source. For PBAP, we used observed levels of arabitol, mannitol, glucose, and trehalose ($\Sigma_{PBAP-Tracers}$) (Table 1), an OC to $\Sigma_{PBAP-Tracers}$ ratio of 14.6±2.1 derived from Yttri et al. (2021.), and an OM:OC conversion factor of 1.75 (Yttri et al., 2011a) to calculate $OM_{PBAP}$

(unit: µg m$^{-3}$). $OM_{BSOA}$ was estimated based on observed levels of 2-methyltetrols ($\Sigma_{BSOA-tracers}$=2-methylerythritol and 2-methylthreitol) (Tab. 2), an OC to $\Sigma_{BSOA-tracers}$ ratio of 165±18 derived from Yttri et al. (2021) and an OM:OC conversion factor of 2.2 (Turpin and Lim, 2001). Yttri et al. (2021) found that 2-methyltetrols traced local BSOA, which was 30% of total identified BSOA constituents in $PM_{10}$, hence total $OM_{BSOA}$ was adjusted accordingly. EC from fossil fuel sources ($EC_{FF}$) was calculated according to eq.9:


$[EC_{FF}] = EC - [EC_{BB}]$                        (eq.9)

where $EC_{BB}$ is obtained from equation 6.

OC from fossil fuel sources ($OC_{FF}$) was calculated according to (eq.10):

$[OC_{FF}] = [EC_{FF}] \times (OC/EC)_{FF}$                       (eq. 10)

where the $(OC/EC)_{FF}$ ratio (2.0±0.25) derived from Yttri et al. (2021) include both primary and secondary OC. An OM:OC
conversion factor of 1.2 was applied for $OC_{FF}$ and 1.1 for $EC_{FF}$. As for $BB_{mass}$ we used the index "mass" in $PBAP_{mass}$, $BSOA_{mass}$ and $FF_{mass}$ to impress that the unit is µg m$^{-3}$.

### 2.1.4 Source apportionment of the absorption coefficient into $eBC_{BB}$ and $eBC_{FF}$

We used positive matrix factorisation with a multilinear engine (PMF-ME2) (Canonaco et al., 2013) and SoFI Pro software
(Canonaco et al., 2021) to apportion $eBC_{BB}$ and $eBC_{FF}$ based on observations of the absorption coefficient. PMF yields factor profiles (here the wavelength dependant absorption, see Yttri et al., 2021) and time series of the emission sources. Two-factor





solutions from repeat bootstrapped PMF runs were mapped via Ångström exponents (AAE, calculated from the factor profiles) using factor 1= lowest AAE, factor 2=highest AAE. An averaged 2 factor solution for each site was then determined, with AAE factor 1; factor 2 of 0.94; 2.01 at Birkenes and 0.8; 1.7 at Zeppelin. Noting that traffic emissions may be partly biofuel

derived, and that residential coal combustion may be partly responsible for the high AAE factor, we identify factor 1 as fossil/liquid fuel eBC and factor 2 as biomass burning/solid fuel eBC consistent with literature AAEs for these sources (Sandradewi et al., 2008, Zotter et al., 2017).

## 2.2 Satellite observations

The Ocean and Land Colour Instrument (OLCI) on board the Sentinel-3 satellites (Donlon et al., 2012), measures the solar radiation reflected by the Earth's surface and atmosphere in 21 spectral bands from the visible to the near infrared. Over land and within 300 km of charted land, the ground spatial resolution is 300 m. The primary objective of OLCI products "[1]. We use OLCI measurements to visualize the aerosol when passing over central Norway and Sweden and to investigate the effect of the aerosol on the measured OLCI radiances by comparison to radiative transfer simulations.

Furthermore, we use observations from the Cloud-Aerosol Lidar with Orthogonal Polarization (CALIOP) on board of the Cloud-Aerosol Lidar and Infrared Pathfinder Satellite Observations (CALIPSO) platform. CALIPSO was launched April 2006 (Winker et al., 2009). CALIOP is a two-wavelength (1064 and 532 nm), polarization-sensitive (at 532 nm) elastic backscatter lidar, which provides globally day- and nighttime profiles of aerosol backscatter, extinction (with an extinction-to-backscatter a priori), and linear particle depolarization with altitude resolution between 30 m and 300 m, below 8.3 km and between 30.1

and 40.0 km, respectively. CALIOP has a small horizontal footprint of 335 m and a revisit time of ~16 days. Here, we utilize the level 2 data products (version 4.21) of the aerosol extinction at 532 nm to evaluate the representation of the dust and BC (from BB) plume in our atmospheric transport simulations. It is given at a spatial resolution of 60 m vertically and 5 km horizontally. The V4 level 2 cloud–aerosol discrimination (CAD) algorithm distinguishes between following tropospheric aerosol subtypes: clean marine, polluted continental/smoke, clean continental, polluted dust, elevated smoke, and dusty marine

(Kim et al., 2018). The data were downloaded from the ICARE Data and Services Center (http://www.icare.univ-lille1.fr/, last access: 19 January 2021).

## 2.3 Radiative transfer model

To understand the processes influencing the OLCI radiances, radiative transfer simulations were made using the DISORT

model (Stamnes et al., 1988; Buras et al., 2011) within the libRadtran framework (Emde et al., 2016). The Subarctic summer

---

[1] https://sentinel.esa.int/web/sentinel/user-guides/sentinel-3-olci/overview





atmosphere (Anderson et al., 1986) was adopted as the ambient atmosphere. Visible Infrared Imaging Radiometer Suite (VIIRS) measurements indicate cloud top height between about 2-4 km and cloud optical depth between 50-100 (worldview.earthdata.nasa.gov). Hence, for the cloudy simulations, a cloud layer of optical depth 70 was included between 2 and 3 km. For simulations over water the Cox and Munk (1954a, b) ocean bidirectional reflectance distribution function

(BRDF) was adopted. Aerosols were included using a profile based on FLEXPART simulation results over Norway (section 3.1). The aerosol optical properties were prepared for input to libRadtran with the MOPSMAP tool (Gasteiger and Wiegner, 2018).

## 2.4 Atmospheric transport models and emission datasets

With the Lagrangian particle dispersion model FLEXPART version 10.4 (Pisso et al., 2019) we model mineral dust in forward
and BC in both forward and backward mode. FLEXPART calculates trajectories of particles to describe transport and diffusion of tracers in the atmosphere. Particles are assumed to be spherical and influenced by gravitational settling, dry deposition and in-cloud and below-cloud scavenging (Grythe et al., 2017). The model is widely applied for LRT modelling of fire emissions (Evangeliou et al., 2016; 2019) and dust sources (e.g. Sodemann et al., 2015).

Emissions from BB were adopted from Copernicus Atmosphere Monitoring Services (CAMS) Global Fire Assimilation
System (GFAS). CAMS GFAS assimilates fire radiative power (FRP) observations from satellite-based sensors converting the energy released during fire combustion into gases and aerosol daily fluxes (Di Giuseppe et al., 2016;Kaiser et al., 2012). Data are available globally on a regular grid with horizontal resolution of 0.1 degrees from 2003 to present. FRP observations assimilated in GFAS are the NASA Terra MODIS and Aqua MODIS active fire products (http://modis-fire.umd.edu/, (Kaufman et al., 2003)). FRP measures the heat power emitted by fires, as a result of the combustion process and is directly
related to the total biomass combusted (Wooster et al., 2005). Using land-use dependent conversion factors, GFAS converts FRP into emission estimates of 44 smoke constituents, such as CO, $CO_2$, $CH_4$, and black-carbon and organic matter components of the aerosol (Kaiser et al., 2012). Here, we used emissions of BC that were subsequently ingested into FLEXPART, which simulated it forward to track atmospheric LRT. The simulations were driven by operational meteorological data from European Centre for Medium-range Weather Forecast (ECMWF) of 1-degree spatial resolution and 3-hourly
temporal resolution. The spatial resolution of the output was set to 0.5 degrees and the temporal resolution to daily. The simulations accounted for wet and dry deposition, assuming a particle density of 1500 kg m$^{-3}$ and a logarithmic size distribution with an aerodynamic mean diameter of 0.25 µm and a standard deviation of 0.3 (Hu et al., 2018;Long et al., 2013) .

Besides forward simulations of BC, we also made backward simulations based on 3-hourly releases from the rural background and remote stations to obtain the emission sensitivity and distinguish sources contributing to BC concentrations at these
locations. We thereby assumed the same properties for BC as in the forward simulation. Emission sensitivity in the bottom 500 m was linked to fields of BC emissions from biomass burning based on GFAS and BC emissions from fossil fuels retrieved from the Evaluating the Climate and Air Quality Impacts of Short-Lived Pollutants (ECLIPSE) emission data set (Stohl et al., 2015; Klimont et al., 2017) on 0.5-degrees resolution.





Emissions of mineral dust are calculated with the FLEXDUST module (Groot Zwaaftink et al., 2016). This module describes
dust mobilization and emission as a function of (threshold) friction velocity following the approach introduced by Marticorena
and Bergametti (1995). Modelled threshold friction velocity is influenced by soil moisture (Fécan et al., 1999) and sediment
regions were identified based on large-scale topography (Ginoux et al., 2001). Emissions are calculated at 0.25 degrees
resolution and 3-hourly interval. The forward simulations include ten size bins for dust smaller than 20 micrometres. For
comparison to measurements based on $PM_{10}$ we will here only consider the size bins with dust up to 10 micrometres. Based
on FLEXDUST emissions, we run two forward simulations of atmospheric transport of dust with FLEXPART. The main
simulation included global dust emissions, the additional simulation included only dust from the Central Asian desert region,
here defined as a square region extending from 42°E to 82°E and 35°N to 50°N. FLEXPART and FLEXDUST simulations of
mineral dust were driven with the same meteorological forcing data as for biomass burning.

For comparison, we will also include estimates from an operational air quality forecast system. The CAMS regional ensemble
forecast (Marécal et al., 2015) is composed of nine air quality models run over a European domain (30°W to 75°E, and 25°N
to 75°N). Forecasts are produced daily and run for 72 hours. The ensemble is taken as the median of the nine models, which
has higher skill than any of individual nine models. The dust product is the median of the prognostic simulations of mineral
dust from each model. Mineral dust is represented in each model by differing size bins and physics, so the dust concentration
represents PM of all sizes associated with mineral dust, which may include size bins up to $PM_{20}$. While each model uses its
own schemes to represent the dust emissions, they all use the dust from the CAMS global model to represent initial and
boundary conditions (Collin, 2021). The CAMS global model is run using C-IFS (Rémy et al., 2019) which simulates dust
emissions, uses three size bins (0.03-0.55 µm, 0.55-0.9 µm, and 0.9-20 µm), and performs assimilation of satellite AOD to
update aerosol concentrations. The CAMS model data were downloaded from the Copernicus Atmospheric Data Store
(https://ads.atmosphere.copernicus.eu/, last access: 18 March, 2021).

**3. Results and discussions**

Although the episode was initially mostly detected based on strong impacts on air quality at the surface, the aerosol was not
only present near the surface. We will first use satellite observations to assess the vertical extent and origin of the plumes. In
Figure 2, a RGB composite of OLCI observations made 2 Oct 2020 is shown. First, the image clearly shows the presence of
aerosols above the cloud layer over Norway/Sweden. Aerosols were thus present, in considerable amounts, at an altitude of at
least 2-3 km. We will further use the OLCI measurements in combination with radiative transfer simulations to determine the
absorption and scattering properties of the aerosol. Four points are marked in Figure 2, indicating cloud only (Cloud), aerosol
over cloud (Aerosol A), aerosol over water (Aerosol B) and water without visible cloud and aerosol (Water). For these four
cases OLCI measurements were averaged over 9×9km² and the averaged spectra and their standard deviations are shown as
solid lines in Figure 3.



The aerosol reduces the radiance by about a factor of 2 when above the cloud layer due to increased absorption by the aerosol compared to the non-absorbing cloud (compare Aerosol A and Cloud in Fig. 3), while above water the aerosol increases the radiance due to increased backscattering (compare Aerosol B and Water in Fig. 3).

To elucidate the aerosol type(s) that may reproduce the OLCI radiances, radiative transfer simulations were made with a combination of highly scattering and highly absorbing aerosols. The scattering aerosols have a single scattering albedo

(SSA) of 0.98 at 400 nm and it decreases to about 0.93 at 1000 nm. The absorbing aerosols have a SSA of 0.28 at 400 nm and it decreases to about 0.1 at 1000 nm. The amounts of scattering and absorbing aerosols were determined by scaling the scattering and absorption aerosol optical depths to get a best match between the OLCI radiances and the simulations, solid and dotted lines in Figure 3 respectively. No single aerosol type was able to reproduce the measurements, rather various combinations of the highly scattering and highly absorbing aerosols were needed to match the measurements. For the water

case (blue lines in Fig. 3), a highly scattering optical depth (OD) of 0.45 and highly absorbing OD of 0.2, both at 550 nm, were used. For Aerosol B (clearly visible aerosol over water, red lines in Fig. 3), highly scattering OD=2.5 and highly absorbing OD=0.7, while for Aerosol A (aerosol over cloud, black lines in Fig. 3) highly scattering OD=2.1 and highly absorbing OD=0.7. Thus, the reproduction of the satellite measurements indicates that the aerosol had an absorbing component.

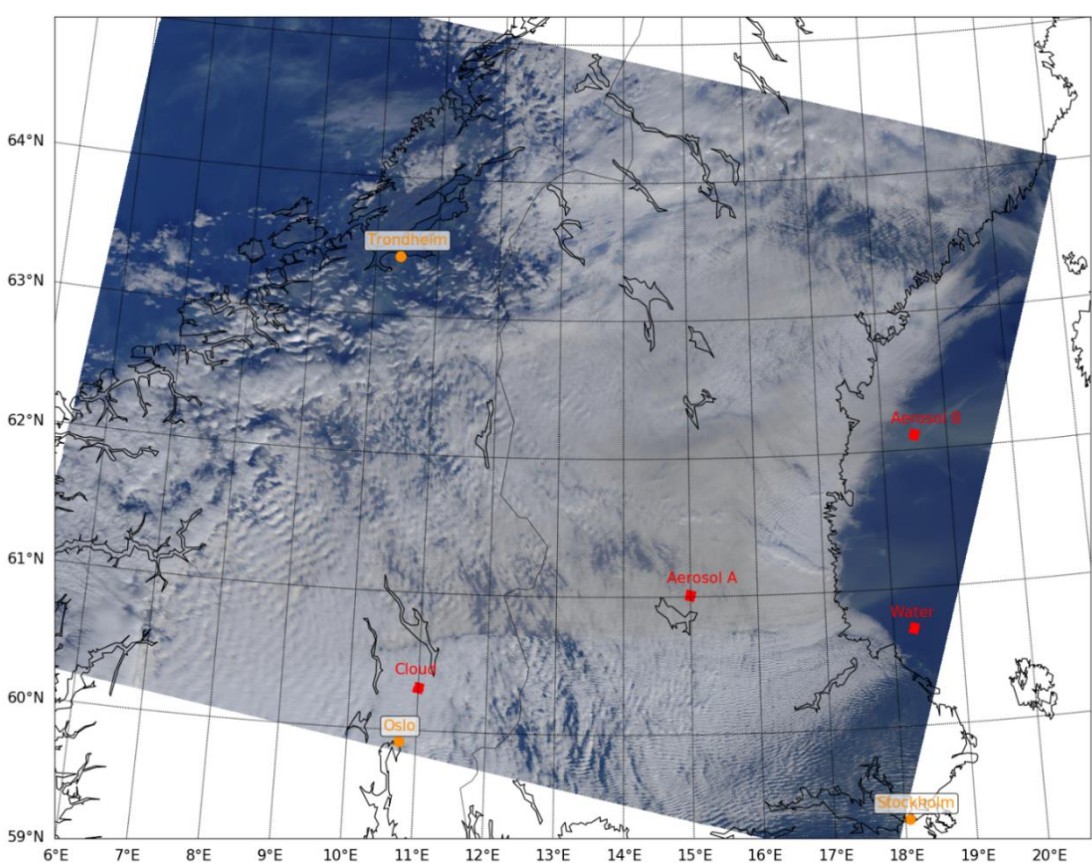

**Figure 2 RGB from OLCI bands 3, 6 and 8 centred at 442.5, 560 and 665 nm. Data from 2 Oct 2020, 09:33 UTC.**





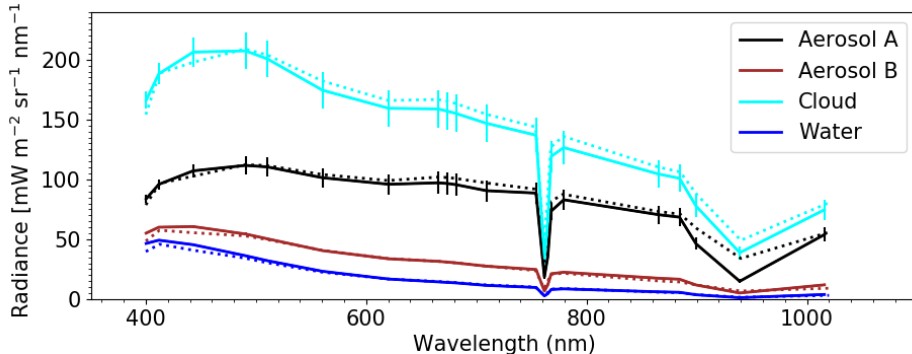

**Figure 3 The OLCI radiances (solid lines) at the four locations marked in Fig. 2. The error bars are standard deviation of the OLCI radiances within the square markers in Fig. 2. Note that for Aerosol B and Water the error bars are small and thus not shown. The dotted lines are radiative transfer model simulations of the OLCI measurements. See section 2.3 for further details.**

The episode was clearly visible from a satellite perspective and the impact on the near-surface air quality was considerable. 24-hour mean $PM_{10}$ concentrations on 2 and 3 October ranged from 8 to 97 µg m$^{-3}$ at Norwegian urban sites, with a median value of 59 µg m$^{-3}$. The episode thereby caused violation of the 24-hour $PM_{10}$ EU-limit of 50 µg m$^{-3}$ at 39 out of 48 Norwegian sites on 2 and/or 3 October, corresponding to 70 days in total. In comparison to years 2018 and 2019, for a selection of 30 sites with measurements in those years, this means that the number of exceedance days during the episode

corresponded to 18% of the average for 2018 and 2019. The number of exceedance days is used to determine whether measures to improve air quality need to be taken, and it does not distinguish whether the cause of poor air quality is due to LRT or local pollutions.

During this episode, it appears that local sources were of minor importance, since 24-hour mean concentrations observed at the rural background site Birkenes, of 66 µg m$^{-3}$ and 61 µg m$^{-3}$, were in the same range as the urban sites. Weekly

mean concentrations at the three rural background sites ranged from 18.6-22.2 µg m$^{-3}$ for $PM_{10}$ and from 4.1-16.1 µg m$^{-3}$ for $PM_{2.5}$, being new maximum values for September–November 2010/11-2019 for all but $PM_{2.5}$ at Birkenes (Fig. 4; Table 1). Reconstructed $PM_{10}$ mass concentration at Zeppelin amounted to 6.4 µg m$^{-3}$. A prevailing coarse fraction ($PM_{10-2.5}$) of $PM_{10}$ was seen at Birkenes (78% of $PM_{10}$) and Hurdal (61%), whereas it was minor at Kårvatn (23%). The spatial variability is not easily explained but was seen at Norwegian urban sites during the episode as well (27-78% in $PM_{10-2.5}$). A pronounced and

even dominating coarse fraction can occur when sources such as sea salt aerosol and mineral dust are prominent.

We thus already see indications of influences from mineral dust and BB aerosols, which can be confirmed by chemical speciation of PM, available at the three rural background sites and the Zeppelin Observatory (Fig. 4; Table 1). For a selection of species, we also show long-term measurements for comparison. With the chemical analysis performed (Table 1), and with assumptions made regarding mineral dust composition (Sect. 2.1.2) and OC and EC conversion factors (Sect. 2.1.3), we were

able to explain 93% (Birkenes), 73% (Hurdal), and 81% (Kårvatn) of the $PM_{10}$ mass at the rural sites. Note that secondary inorganic aerosol constituents and inorganic anions and cations were not available for the Kårvatn site.





**Figure 4** Panels show concentrations observed in the October 2020 episode (red diamonds: Birkenes, orange circles: Hurdal, purple triangles: Kårvatn, blue diamonds: Zeppelin) along with box plots (5, 25, 50, 75, 95 percentiles and outliers) at the rural background sites Birkenes, Hurdal and Kårvatn and remote site Zeppelin of weekly data in the period Sept.-Nov. for a) $PM_{10}$ mass concentration (2010-2019); b) $PM_{2.5}$ mass concentration (2010-2019); c) Organic Carbon (OC) in $PM_{10}$ (2010-2019 for Birkenes, 2011-2019 for Hurdal and Kårvatn 2017-2019 for Zeppelin); d) OC in $PM_{2.5}$ (similar as for c except Zeppelin with no measurements); e) Elemental Carbon (EC) in $PM_{10}$ (similar as for c); f) EC in $PM_{2.5}$ (similar as for d); g) Crustal elements in $PM_{10}$ (2014-2019); h) Levoglucosan (2010-2019), mannitol, and 2-methylerythritol in $PM_{10}$ (2016-2018); i) Crustal elements in $PM_{10}$ at Zeppelin (2018-2019); j) Levoglucosan , mannitol, and 2-methylerythritol in $PM_{10}$ at Zeppelin (2017-2019). In panel g) and h) the box plots are for Birkenes only.



## 3.1 Mineral dust

Mineral dust elements (Al, Fe, Mn, and Ti) were all highly elevated during the episode. At Birkenes, the Al and Fe levels were over 30 times higher than the long-term mean (2016-2019), whereas the corresponding factors for Ti and Mn were
respectively 27 and 16 (Table 1; Fig. 4). Maximum ever concentrations were observed for all mineral dust elements at Birkenes by a fair margin, suggesting that an event of this magnitude is rare. Note that long term time series of mineral dust are available only for Birkenes. Kårvatn experienced the highest concentration for all the mineral dust elements during the episode, followed by Birkenes and Hurdal, although with a minor difference. The relative composition of the mineral dust elements was indistinguishable between the three rural background sites and the Zeppelin Observatory. Al was most abundant at all sites
and concentrations of other elements relative to Al are similar at all sites: Al:Fe:Ti:Mn =1:(1.5-1.6):(26-38):(56-65). This similar relative composition points at a common origin of the observed mineral dust. The Al:Fe ratio points to Eurasian mineral dust sources (Crocchianti et al., 2021), which is supported by our FLEXPART/FLEXDUST model simulations and CAMS regional mineral dust product, discussed below. Observed elements at Zeppelin were of similar composition yet observed masses were 2.4 – 3.7 times lower than the mean at the rural background sites, which is not surprising given the larger distance
to the source region.

**Table 1: Mean concentrations of PM and associated species observed during the episode compared to the long-term weekly mean (±SD) for Sept.-Nov. (2010/11-2019) for Birkenes, Hurdal, Kårvatn and Zeppelin, except crustal elements at Birkenes (2014-2019) and Zeppelin (2018-2019), EC/OC and organic tracers at Zeppelin (2017-2019).**

| | Birkenes Observatory | | Hurdal | | Kårvatn | | Zeppelin Observatory | |
|---|---|---|---|---|---|---|---|---|
| *Mass concentration (µg m⁻³)* | Episode | Mean (±SD) | Episode | Mean (±SD) | Episode | Mean (±SD) | Episode | Mean (±SD) |
| PM$_{10}$ | 18.7 | 4.9±3.4 | 22.2 | 4.3±2.6 | 21.0 | 2.4±2.1 | 6.4[1) ] | N/A |
| PM$_{2.5}$ | 4.1 | 2.5±2.2 | 8.6 | 2.4±1.6 | 16.1 | 1.4±1.3 | N/A | N/A |
| PM$_{10-2.5}$ | 14.6 | 2.6±2.0 | 13.6 | 2.0±1.4 | 4.9 | 1.1±1.0 | N/A | N/A |
| *Carbonaceous aerosol (µg C m⁻³) PM$_{10}$* | | | | | | | | |
| OC | 2.0 | 0.72±0.49 | 2.9 | 1.2±0.77 | 3.3 | 0.72±0.61 | 0.81 | 0.051±0.079 |
| EC | 0.41 | 0.09±0.07 | 0.38 | 0.13±0.06 | 0.37 | 0.05±0.04 | 0.07 | 0.006±0.010 |
| CO$_3^{2-}$ | 0.05 | N/A | 0.08 | N/A | 0.11 | N/A | 0.02 | N/A |



|  | Birkenes | Observatory | Hurdal |  | Kårvatn |  | Zeppelin | Observatory |
|---|---|---|---|---|---|---|---|---|
| *PM$_{2.5}$* |  |  |  |  |  |  |  |  |
| OC | 0.67 | 0.48±0.33 | 1.5 | 0.59±0.30 | 2.4 | 0.42±0.29 | N/A | N/A |
| EC | 0.20 | 0.09±0.07 | 0.30 | 0.12±0.06 | 0.32 | 0.05±0.04 | N/A | N/A |
| CO$_3^{2-}$ | 0.03 | N/A | 0.06 | N/A | 0.09 | N/A | N/A | N/A |
| *PM$_{10-2.5}$* |  |  |  |  |  |  |  |  |
| OC | 1.3 | 0.25±0.24 | 1.3 | 0.67±0.68 | 0.93 | 0.26±0.28 | N/A | N/A |
| EC | 0.20 |  | 0.08 |  | 0.06 |  | N/A | N/A |
| CO$_3^{2-}$ | 0.02 | N/A | 0.02 | N/A | 0.02 | N/A | N/A | N/A |
| *Secondary inorganic aerosol (µg m$^{-3}$)* |  |  |  |  |  |  |  |  |
| SO$_4^{2-}$ | 1.6 | 0.80±0.77 | 1.5 | 0.50±0.46 | N/A | 0.27±0.37 | 0.44 | 0.25±0.26 |
| NO$_3^-$ | 1.7 | 0.90±0.98 | 0.71 | 0.54±0.73 | N/A | 0.24±0.43 | 0.23 | 0.20±0.38 |
| NH$_4^+$ | 0.60 | 0.26±0.35 | 0.39 | 0.19±0.25 | N/A | 0.10±0.21 | 0.037 | 0.039±0.075 |
| *Inorganic anions and cations (ng m$^{-3}$)* |  |  |  |  |  |  |  |  |
| Ca$^{2+}$ | 500 | 40.8±29.2 | 540 | 28.8±21.6 | N/A | 23.7±20.1 | 201 | 43.7±59.2 |
| Mg$^{2+}$ | 122 | 61.8±44.5 | 58.5 | 19.1±13.5 | N/A | 15.1±11.1 | 35 | 36.2±26.6 |
| Na$^+$ | 583 | 501±371 | 85.9 | 136±113 | N/A | 102±98.7 | 80 | 239±178 |
| K$^+$ | 170 | 57.4±36.7 | 147 | 49.3±28.6 | N/A | 27.4±20.2 | 160 | 22.8±23.6 |
| Cl$^-$ | 666 | 627±538 | 81.2 | 115±119 | N/A | 123±149 | 114 | 339±291 |
| *Crustal elements (ng m$^{-3}$)* |  |  |  |  |  |  |  |  |
| Al | 819 | 24.7±16.2 | 734 | N/A | 1000 | N/A | 254 | 53.8±36.6 |
| Fe | 540 | 16.5±16.6 | 485 | N/A | 650 | N/A | 162 | 26.1±17.5 |
| Ti | 21.5 | 0.80±0.67 | 19.6 | N/A | 28.1 | N/A | 9.7 | 1.65±0.1.11 |





| | Birkenes Observatory | | Hurdal | | Kårvatn | | Zeppelin Observatory | |
|---|---|---|---|---|---|---|---|---|
| Mn | 14.3 | 0.89±0.83 | 13.1 | N/A | 16.6 | N/A | 3.9 | 0.45±0.29 |
| **Organic tracers** | | | | | | | | |
| *(ng m⁻³)* | | | | | | | | |
| *Biomass burning* | | | | | | | | |
| Levoglucosan | 28.5 | 10.6±11.3 | 27.8 | N/A | 33.9 | N/A | 5.0 | 0.53±0.65 |
| Mannosan | 5.92 | 2.04±2.63 | 6.66 | N/A | 5.48 | N/A | 0.98 | 0.082±0.104 |
| Galactosan | 1.03 | 0.52±0.66 | 1.79 | N/A | 1.60 | N/A | 0.27 | 0.024±0.028 |
| *Biogenic Secondary Organic Aerosol* | | | | | | | | |
| 2-methylerythritol | 0.667 | 0.20±0.34 | 0.551 | N/A | 0.349 | N/A | 0.105 | 0.085±0.199 |
| 2-methylthreitol | 0.242 | 0.081±0.128 | 0.267 | N/A | 0.122 | N/A | 0.055 | 0.042±0.086 |
| *Primary Biological Aerosol Particles* | | | | | | | | |
| Arabitol | 16.0 | 6.57±6.30 | 17.9 | N/A | 34.6 | N/A | 0.66 | 0.14±0.30 |
| Mannitol | 16.0 | 7.08±6.60 | 18.1 | N/A | 34.7 | N/A | 0.82 | 0.16±0.34 |
| Glucose | 11.7 | 5.32±4.47 | 9.00 | N/A | 13.6 | N/A | 1.61 | 0.37±0.47 |
| Trehalose | 9.15 | 4.58±4.94 | 8.94 | N/A | 22.2 | N/A | 1.27 | 0.21±0.41 |

1) Based on reconstructed mass

We find a lower estimate of the mineral dust concentration at the rural background sites ranging from 5.6-7.6 µg m⁻³ (eq.2) and an upper estimate ranging from 6.9-8.6 µg m⁻³ (eq.3), whereas 1.9-2.5 µg m⁻³ was attributed to mineral dust at the remote site. Only $CO_3^{2-}$ was included in the upper estimate for Kårvatn, as data needed to calculate the oxides was missing. The oxides

increased the upper estimate at Birkenes and Hurdal by 10%. If we assume a similar increase at Kårvatn, the upper estimate increases from 8.6 µg m⁻³ to 9.4 µg m⁻³. The lower estimate provides a 25-36% contribution to $PM_{10}$ at the rural background sites whereas the upper estimate ranges from 31-45%. Similarly, at Zeppelin, we found that 32-41% of $PM_{10}$ was mineral dust. Note though, that we do not have measurements of $PM_{10}$ concentrations at Zeppelin. We therefore assumed that our PM reconstruction from different constituents (sections 2.1.2 and 2.1.3) explains as much of the total $PM_{10}$ as it did for the

background sites Birkenes and Hurdal, which is on average 82%.





Simulations with FLEXPART, based on dust emissions from FLEXDUST, help us to further demonstrate and understand the LRT of mineral dust during this episode. Figure 5 shows the modelled surface concentrations on 27 September, during the storm in Central Asia that caused the large mineral dust emissions, and on 2 October. These time steps were chosen based on the availability of CALIOP overpasses that captured the dust (or BC) plume, which will be discussed later.

FLEXPART results show a distinct dust plume transporting dust from the regions east of the Caspian Sea towards north-west Europe. The dust plume partly overlaps with a wildfire plume starting in Ukraine, as is illustrated by the black lines in the right hand side figures. The contour lines of 500hPa geopotential height illustrate how transport of mineral dust originating from Central Asia is forced on a front between a high-pressure region over Russia and low pressure in Europe. We see enhanced mineral dust concentrations over North Africa and a plume of dust transported over eastern Europe. Dust originating from

North Africa is partly mixed into the dust plume from Central Asia yet contributes little to the increased surface concentrations of mineral dust observed in Norway during this episode, as will be further discussed below. Natural dust sources in Central Asia include, e.g., the Karakum and Aralkum deserts in Turkmenistan and Kazakhstan. These deserts are part of what is sometimes referred to as the dust belt, extending from the west coast of Africa to China (Prospero et al., 2002). There is a variety of deserts in Central Asia with different characteristics. Most of these dust sources are active between March and

October (Shen et al., 2016; Indoitu et al., 2012). According to our simulations, a dust storm occurred in Central Asia in the end of September. From the 25th of September total dust emissions in this region started increasing, reaching maximum values on the 27th of September, and slowly decreasing again until the 3rd of October. Dust emissions were up to 2 g m$^{-2}$ h$^{-1}$ and total emissions from this region amounted to 5.1 Tg in 8 days.

Figure 6 shows the curtain of aerosol extinction profiles at 532 nm (colour coded, the overpass is plotted in Fig 5.) with

FLEXPART mineral dust and BC concentration contour lines overlayed. The top panel shows CALIOP aerosol extinction from 26 Sept., time of first record: 23:48:09, between 36° and 45°E, with dominating aerosol subtype (not shown here) dust and polluted dust, at the surface around 40.5° and 43°E, elsewhere and at elevated levels. The FLEXPART simulation captured both the elevated concentrations near the surface and the vertical spread of the dust plume at the time of emissions and near the source region of the dust (white contours). The dust plumes reached Norway on 2 October. Figure 6 shows CALIOP aerosol

extinction from that day, at around midnight (middle panel, 13.5-29°E) and around noon (lower panel, 0-9°E). The modelled dust plumes partly coincide with the satellite images. The night-time CALIOP curtain shows an area with enhanced aerosol loading extending to above 5 km of polluted continental/dust west of 16°E, and a lower region with dust/polluted dust east of 20°E. Also note that the lack of absorbing aerosols around 20°E in the satellite images is locally due to obstructions by clouds. Twelve hours later the CALIOP overpass the lowermost tip of Norway, showing mineral dust reaches over 4 km height (around

5°E). An animation of dust RGB (Red, Green, Blue) composite images, which are based on infrared channel data from the Spinning Enhanced Visible and Infrared Imager (SEVIRI) is provided as a supplement. Regions with bright pink colours[2]

---

[2] See https://www-cdn.eumetsat.int/files/2020-04/pdf_rgb_quick_guide_dust.pdf



visualises the dust transport over several days (September 30 - October 3). Images were downloaded from EUMETSAT (https://eumetview.eumetsat.int/mapviewer/?product=EO:EUM:DAT:MSG:DUST).

**Figure 5 Modelled surface concentrations of BC from biomass burning (left) and mineral dust (right) on 27 September and 2 October (midnight and noon). Blue contours: ECMWF 500 hPa geopotential height (m). In the right column black contour lines indicate the regions of the modelled wildfire plume (BC>0.03 µg m$^{-3}$) as shown in the left column. BC simulations started only after 27 September because no relevant emissions were observed for the episode in Norway. The red lines indicate the location of CALIOP overpasses shown in Fig. 6.**

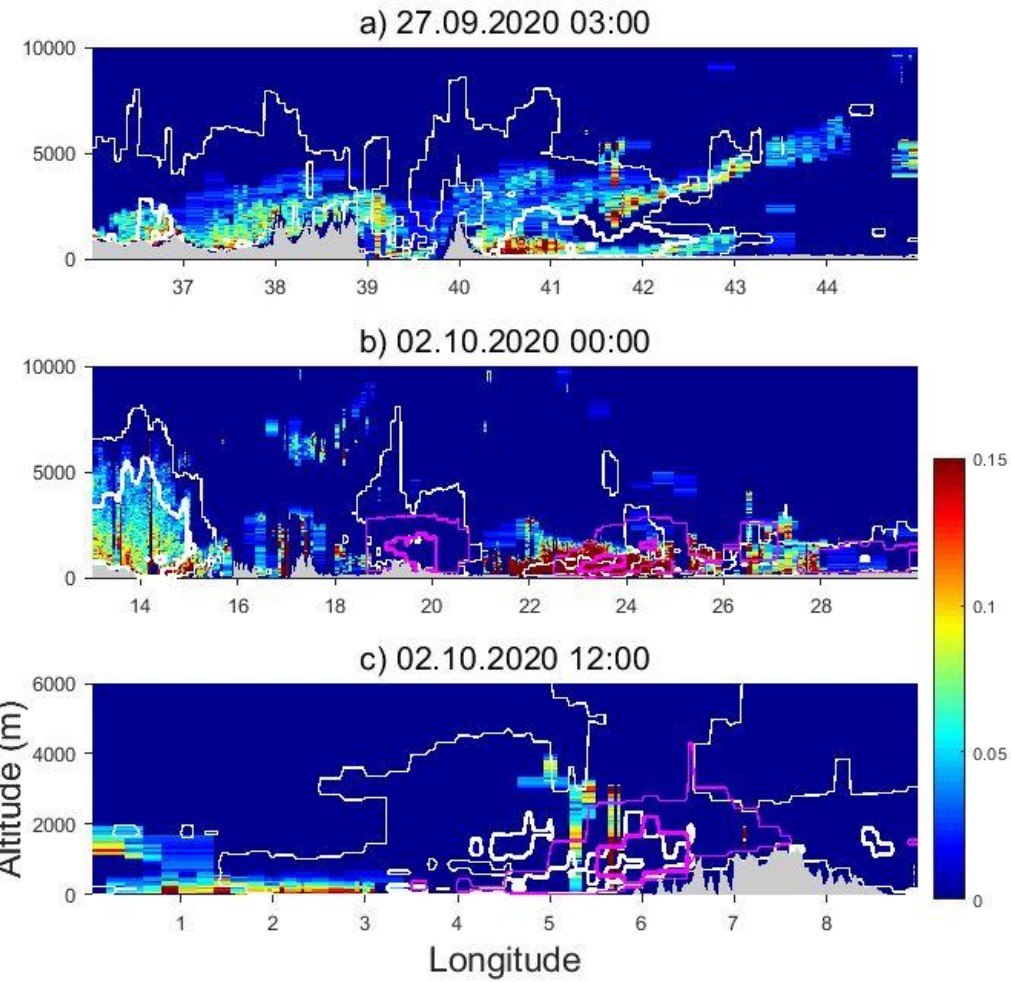


**Figure 6 Aerosol extinction at 532 nm [units: km-1] during three overpasses (see Fig. 5 for location) is shown as colour plot (a. CALIOP overpass from 26 Sept., time of first record 23:48, b: 2. October 01:18, and c. 2 October 11:55). The titles of panels a-b indicate the FLEXPART 3-hourly time step corresponding to the overpass. White contours show FLEXPART mineral dust concentrations (thick and thin lines, top: 40 and 6 µg m-3, middle: 50 and 10 µg m-3, bottom: 15 and 3 µg m-3). Magenta contours show FLEXPART BC concentrations (thick and thin lines, top: none, middle: 50 and 5 ng m-3, bottom: 40 and 10 ng m-3).**

We compare time series of the FLEXPART modelled surface concentrations of mineral dust to the observed $PM_{10}$ concentrations at Birkenes in Figure 7. (Time series of $PM_{10}$ are not available at the other stations included in the model results.) We further show surface concentrations from the CAMS regional model product over Europe, which includes the three stations in Norway but not the remote station in the Arctic. Both FLEXPART and CAMS regional show increased surface concentrations of mineral dust at all three stations between 28 and 30 September and 1 and 4 October. While the first peak is not seen clearly in the observation the second peak is visible in both models for all stations. The timing of the second peak value appears similar, yet continuation of the event is different. Moreover, the estimated values of surface concentrations are



larger in FLEXPART than in CAMS regional. This is even more obvious for the weekly averages shown in the bottom panel of Figure 7, where FLEXPART modelled concentration are a factor of 1.7 to 3.4 higher than CAMS. The lower estimates of
mineral dust by CAMS are probably a result of various factors. One issue was apparent when we examined the dust concentrations from each of the nine ensemble members. At Birkenes all models and at Kårvatn all except for one of the models underestimated the dust concentration. We also found that the EURAD and MOCAGE models tended to greatly underestimate the dust concentrations (values ranging from $0.1 - 3.1$ µg m$^{-3}$), which will contribute to an overall lowering of the median. For the MOCAGE model, this has been linked to the deposition of desert dust being too high and an incorrect
treatment of the boundary conditions from the CAMS global model in a previous MOCAGE model cycle and both issues have been subsequently corrected (Mathieu Joly, personal comm.). The dust plume from Central Asia did not reach Zeppelin until 4 October and peak surface concentrations are lower than for the rural stations, complying with the observations. Comparison to average dust concentrations based on 8-10 days sample measurements (Figure 7, bottom) indicates a nice agreement between the model and observations, although modelled dust concentrations are overestimated at Zeppelin. From an additional
simulation that only included mineral dust emissions from Central Asia, we find that desert regions in Central Asia are major contributors to the dust episode, accounting for 88% of surface dust concentrations at the three rural background sites. The modelled relative contribution of mineral dust from Central Asia is similar at all sites, pointing at common sources. This agrees with the observed relative composition of the mineral dust elements between the three sites being indistinguishable. At Zeppelin, the model finds a relatively larger contribution from other, most likely local, sources. No indication of this was seen
in the observations, suggesting that local mineral dust emissions were overestimated by the model.





**Figure 7 PM₁₀ concentrations observed at Birkenes (top). Mineral dust surface concentrations as simulated with FLEXPART and CAMS-regional at three stations in Norway and the remote station (outside CAMS domain). Bottompanel; mean concentrations from simulations compared to observations from weekly samples for Birkenes (30 September – 7 October 2020), Hurdal and Kårvatn (28 September – 5 October 2020) and Zeppelin (2 – 12 October 2020). For FLEXPART estimates in the bottom panel light green colours indicate dust from global sources and dark green colours indicate the contribution of dust from Central Asia only.**



### 3.2 Biomass burning

We calculated that carbonaceous aerosol, which generally is the major fraction of biomass burning aerosol, made a 26-36% contribution to $PM_{10}$, 34-44% to $PM_{2.5}$, and 21-43% to $PM_{10-2.5}$ (Table 1) when converting OC and EC to account for other elements than just Carbon. Adding $CO_3^{2-}$, increased the contribution only to a minor extent (1-4%). The carbonaceous aerosol made a similar contribution to reconstructed $PM_{10}$ at Zeppelin (29%) as for the rural background sites, as did $CO_3^{2-}$ (1%). BB aerosol mainly consist of EC and OC. Observed EC levels at the rural background sites were either a record high or top

six, considering both the annual (not shown) and the Sept.-Nov. 2010/11-2019 time series (Table 1). OC (in $PM_{10}$) levels were also highly elevated, i.e., within the 95-99th percentile. The OC level was noticeably higher at Kårvatn and Hurdal compared to Birkenes, whereas EC (in $PM_{10}$) was highest at Birkenes, although by a short margin (Fig. 2). The OC and EC levels observed at Zeppelin were the highest reported since regular measurements started in 2017, but still 3 times lower compared to the major wildfire episode influencing the site in April/May 2006 (Stohl et al., 2007).

The split between the fine and coarse fraction of $PM_{10}$ varied substantially between EC and OC at the rural background sites, and between sites. EC results from incomplete combustion of biomass and fossil fuel and is almost exclusively associated with fine aerosol particles. Hence, one would expect rather high EC values for the fine fraction of $PM_{10}$. The 50:50 percent split between the fine and coarse fraction of $PM_{10}$ seen at Birkenes therefore is a rare finding, whereas the 79:21 percent and 84:16 percent splits for the two other rural background sites are closer to the long-term mean (97:3). OC was even more skewed

towards the coarse fraction than EC, with a fine/coarse split ranging from 34:66 percent at Birkenes to 72:28 percent at Kårvatn. The reason for such an atypically high coarse fraction could be due to condensation, agglomeration, and heterogenic chemical reactions influencing the size distribution of carbonaceous aerosol, and, for OC, contribution of PBAP. Another possibility is that pyro convection entrains large, partly combusted particles from the ground to such high altitudes that they can be subject to LRT. Further, Dusek et al. (2017) and Yttri et al. (2021) pointed to charring of coarse fraction PBAP during thermal-optical

analysis (TOA), forming pyrolytic carbon that is erroneously interpreted as coarse EC. In fact, the episode coincided with the time of the year when PBAP peak (early fall) (Yttri et al., 2007a; Yttri et al., 2021), and the PBAP tracer levels were highly elevated but still comparable to levels previously seen at this time of the year (Fig.4, Table 1). A lower estimate of 12-16% (Eq. 7) was calculated for $BB_{mass}$ to $PM_{10}$ at the rural background sites, whereas the upper estimate ranged from 17-21%. The lower estimate apportioned only 7.6±1.8% of $PM_{10}$ to $BB_{mass}$ at Zeppelin, which likely experienced

a more pronounced degradation of levoglucosan due to its remote location. The upper estimate of 17% is in line with that observed at the rural background sites. Estimates of the major carbonaceous aerosol (OC and EC) sources (BB, BSOA, FF, and PBAP) can be derived from the source specific organic tracers listed in Table 1 (see section 2.1.3 for details). Here we estimated the $PBAP_{mass}$, $BSOA_{mass}$ and $FF_{mass}$ contribution to $PM_{10}$ for a better constraint and understanding of the BB source, finding that that their joint contribution made

an equally large contribution to $PM_{10}$ as the lower estimate of BB at both the rural background sites (13-17%) and at the remote



site (6.4%), underlining the importance of the BB source. Combined, the contribution of $BB_{mass}$ (lower estimate), $PBAP_{mass}$, $BSOA_{mass}$ and $FF_{mass}$ ought to match the observed level of the carbonaceous aerosol (here: Sum of OM and EC×1.1). For the rural background sites this matched quite well (99±21%), whereas it only amounted to 48% at the remote Arctic site. The most likely explanation of this discrepancy is failure to account for degradation of organic tracers, and particularly levoglucosan.

There are indications that 2-methyltetrols have short atmospheric lifetimes as well (Yttri et al., 2021), whereas the low levels of PBAP tracers at Zeppelin likely reflect the scarce vegetation of the Artic biome. Further, the ER ratios used for PBAP and BSOA (sect. 2.1.4) are derived from measurements in the Boreo-Nemoral biome, thus their suitability in the Arctic biome is a matter of discussion. Consequently, the apportionment of $BB_{mass}$, $PBAP_{mass}$, $BSOA_{mass}$ and $FF_{mass}$ is associated with greater uncertainty than for the rural background sites.

High time resolution measurements of eBC at Birkenes was attributed to a biomass burning fraction ($eBC_{BB}$) and a fossil fuel combustion fraction ($eBC_{FF}$), apportioning 43% to $eBC_{BB}$ and 57% to $eBC_{FF}$ for the episode in question (midnight 1-2 October to midnight 3-4 October). Extending the period to match that of the weekly sample (30 September–7 October) reduced the $eBC_{BB}$ fraction to 35%, whereas the $eBC_{FF}$ fraction increased to 65%. The $eBC_{BB}/eBC_{FF}$ split is thus comparable to the levoglucosan approach, which apportioned equally large shares to $EC_{BB}$ and $EC_{FF}$ for 30 September-7 October but note that

the range (50±20%) of the levoglucosan approach is very wide. $eBC_{BB}$ ($R^2$=0.82) correlated higher with the high time resolution measurements of $PM_{10}$ at Birkenes than $eBC_{FF}$ ($R^2$=0.67), suggesting that biomass burning emissions were more important for the evolution of PM than fossil fuel sources. $eBC_{FF}$ explained 60% of eBC at Zeppelin and $eBC_{BB}$ 40%, considering both the episode (2-7 October) and the longer period covered by the filter sample (2-12 October). As for Birkenes, this corresponds with the levoglucosan estimate for EC, which apportions an equally large fraction to $EC_{FF}$ and $EC_{BB}$.

Time series of eBC at Birkenes and Zeppelin are shown in figures Figure 8 and Figure 9. The measurements showed that eBC values from both biomass burning and fossil fuel were of similar magnitude and both reached peak values on 3 October at Birkenes and on 5 October at Zeppelin. Also, modelled black carbon from biomass burning peaks on 3 October at Birkenes, yet simulated values are underestimated compared to observations. BC concentrations at the other background stations are of similar magnitude, while there is a delay in peak concentration at Kårvatn compared to the other stations. The high correlation

($R^2$=0.92) between eBC and the major OM fraction at Zeppelin (Fig. 9), obtained from the collocated Aerosol Chemical Speciation Monitor Time-of-Flight (ACSM-ToF) instrument, points to combustion sources as the origin of OM. The correlation was more pronounced for $eBC_{FF}$ ($R^2$=0.83) than for $eBC_{BB}$ ($R^2$=0.66). The episode of enhanced BC from fossil fuel combustion does not show in our model results for the rural background stations and is strongly underestimated at Zeppelin. Explanations can be found in the use of monthly mean fossil fuel BC emissions that could average out effects of short-term

emissions.





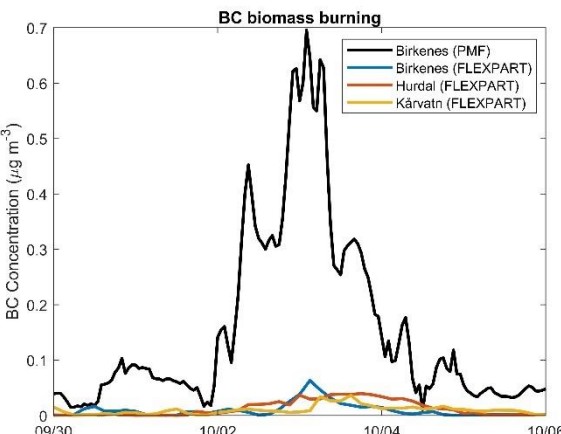

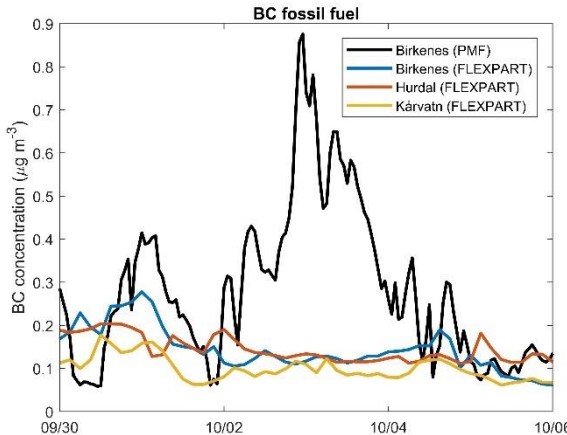

**Figure 8** eBC$_{BB}$ (left) and eBC$_{FF}$ (right) concentrations retrieved with PMF from observations at Birkenes (black) and simulated with FLEXPART at Birkenes, Hurdal and Kårvatn.

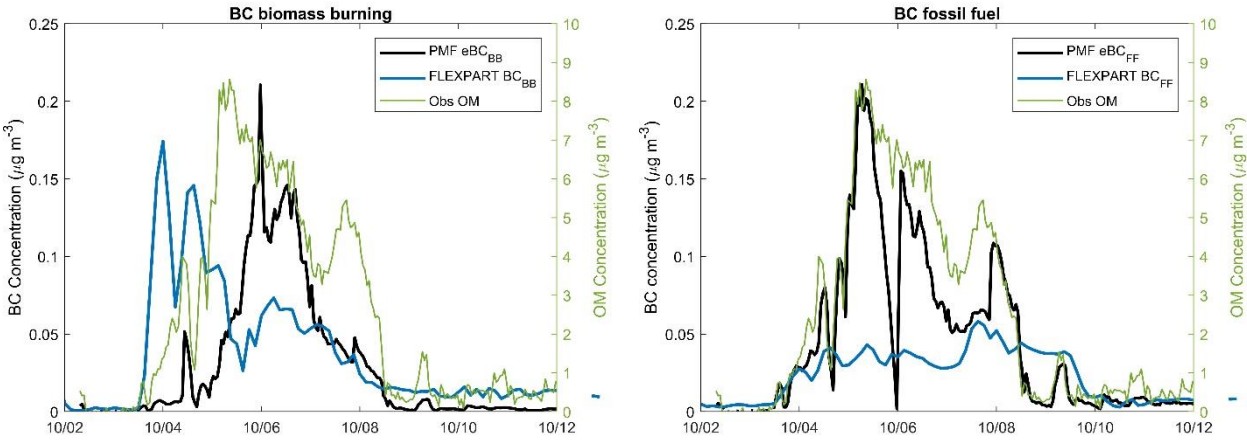

Figure 9 eBC$_{BB}$ (left) and eBC$_{FF}$ (right) concentrations retrieved with PMF from observations (black) and simulated with FLEXPART (blue) and observed OM concentrations (green) at Zeppelin.

We further investigate the LRT of BC from biomass burning based on maps and cross-sections coinciding with CALIOP observations, like for mineral dust (Fig. 6). Although BC from biomass burning is strongly underestimated at Birkenes, the model results do show elevated BC concentrations over southern Norway on 2 October. In the CALIOP profiles, modelled BC plumes (pink contour lines) coincide with regions of very strong aerosol extinction, though being mainly categorized as dust/polluted (22-28°E). While not explicitly shown here, we would like to note, that also the TROPOspheric Monitoring Instrument (TROPOMI) on board of the Copernicus Sentinel-5 Precursor satellite, detects enhanced aerosol index values and increased CO total column over the Baltic countries on October 1-2, which confirms the presence of UV absorbing aerosols,





including biomass burning aerosols, in this region. In the region at approximately 20°E on 2 October 00:00 a clear BC signal is modelled, yet absent in the CALIOP profile due to clouds.

We use backward modelling to characterize the source regions for the observed air pollution. The source -receptor relationship (SRR) for BC at Birkenes during this episode is shown in Figure 10 (top panel). As for the dust plume, long-range transport is strongly influenced by a low- and high-pressure system (Fig. 5). The strong winds from the east confine the SRR mostly to a

region extending between Birkenes and Central Asia. Wildfires contributing to the BC concentrations at Birkenes are thus mostly restricted to this region, and largest contributions are seen in Ukraine and southern Russia, although some contributions from fires in North America and northern Russia are seen as well. Due to the combination of a strong underestimation of modelled BC from biomass burning and good representation of dust concentrations at our stations, we expect that the BC emissions are a larger cause of error than the actual atmospheric transport modelling. It is thus likely that we are missing some

sources contributing to the plume (Figure 10, top panel) or that locations are correct, yet total emissions are underestimated. Fossil fuel emissions are more widespread and mostly emissions in southern Sweden and from the Baltic States down to Ukraine contribute to the BC concentrations at Birkenes (Figure 10, bottom) in this event. Similar results were seen at the stations Kårvatn and Hurdal (not shown). At Zeppelin, there was an enhanced influence of fossil fuel emissions in northern Europe on BC levels.

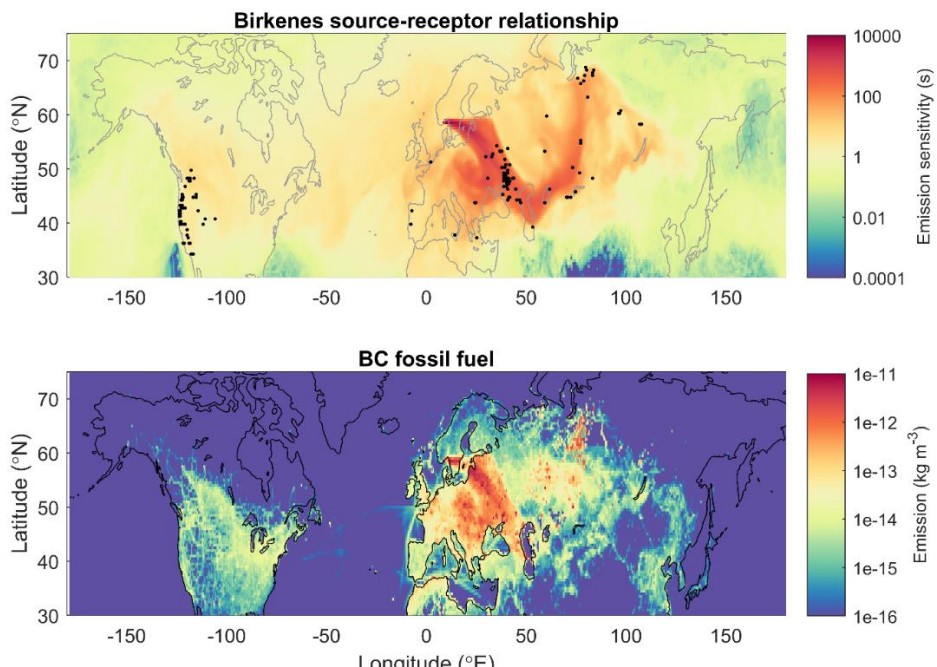


**Figure 10 BC source-receptor relationship at Birkenes (2 October 2020 00:00 – 4 October 2020 00:00) modelled with FLEXPART in backwards mode (top) and BC fossil fuel emissions (bottom). Markers in the top panel indicate locations with BC emissions from biomass burning that contribute 0.1>ng m⁻³ to BC concentrations at Birkenes. Only emissions that contribute to the modelled concentrations at Birkenes are included in the lower panel. The source-receptor relationship includes air masses up to 100 m**

**above ground level.**



## 4. Conclusions

An exceptional episode of elevated $PM_{10}$ concentrations at several measurement sites, with levels exceeding those on any other occasion in the last 10 years, was observed on 2 and 3 October 2020 in Norway, and elsewhere in northern Europe. We have analysed this episode based on surface observations, satellite observations, and atmospheric transport modelling.

Long-range transport of PM was recorded with satellite observations, showing aerosols above the cloud layer. Radiative transport simulations with concentration profiles of different aerosol types were performed to gain an understanding of the processes influencing radiances observed by the OLCI (Sentinel-3). These indicated that there was a contribution from both absorbing and scattering particles for this episode. The rather larger contribution of coarse fraction relative to the fine fraction of $PM_{10}$, however, pointed more in the direction of mineral dust. Chemical analysis of surface samples was thus necessary to

determine the sources. In our surface observations, a clear influence of biomass burning was revealed (12-21%), as well as mineral dust (25-45%).

Surface concentrations of crustal elements Al, Fe, Ti and Mn all strongly exceeded previous maximum recordings. At Birkenes, crustal element levels were 16 to 30 times higher than long-term mean values. Contributions of mineral dust were estimated to be 25-36% of $PM_{10}$ as a minimum and 31-45% as a maximum. The ratios between crustal element levels were similar at all

stations and pointed to a common dust source. Based on atmospheric transport simulations, we concluded that Central Asia, including the Karakum desert, was the main source of mineral dust observed in Norway during this episode, contributing roughly 88% to surface dust concentrations.

Coincidentally, biomass burning emissions in the same transport pathway as the mineral dust plume, caused simultaneous peaks in $PM_{10}$. Contributions of biomass burning to $PM_{10}$ were estimated to be 12-16% up to 17-21% at the rural background

stations and 8-17% at the remote site. Fires in Ukraine were a source of long-range transport to the background stations, as shown with backwards transport modelling. The model results however, underestimated the retrieved BC levels at Birkenes and Zeppelin based on surface eBC observations. It could thus be that emissions in this region were underestimated, or that additional sources were relevant. A qualitative comparison between CALIOP observations and FLEXPART model output, although limited due to cloud coverage, suggested that the model does capture the location of the BC plume.

Our analysis reveals how unrelated emission sources can combine during long range transport to cause extreme adverse air quality events in Norway. Considering the 24-hour $PM_{10}$ EU-limit of 50 µg m$^{-3}$, a total of 70 exceedance days at 39 stations were observed in this single event. It thereby corresponds to 18% of the annual mean number of exceedance days in Norway. This shows the large impact long-range transport episodes may have on air quality regulations. The combined use of remote sensing, high quality measurements and transport modelling proved effective in describing the episode and distinguishing its

causes.



**Data availability**

All in situ data are reported to the EMEP monitoring programme (Tørseth et al., 2012) and are available from the database infrastructure EBAS (http://ebas.nilu.no/) hosted at NILU. Measurement data of PM from Norwegian cities and municipalities are available at: api.nilu.no. FLEXPART simulation results are available from the authors upon request.

**Author contributions**

KEY, WA and MJ provided analysis and interpretation of surface observations. SMP provided eBC PMF analysis. HU provided (historical) elemental data. AK prepared analysis of the OLCI data and radiative transfer modelling. KS and SE prepared visualisation and interpretation of CALIOP data. PH contributed to retrieval and interpretation of CAMS regional dust product. NE, SE and CGZ performed FLEXDUST and FLEPXART simulations and prepared analysis and visualisation

of model results. CGZ and KEY wrote the original draft of the paper. All authors contributed with critical review, commentary, and editing to the manuscript.

**Acknowledgements**

We thank NASA and CNES engineers and scientists for making CALIOP data available. The lidar data were downloaded from the ICARE Data and Service Center. Furthermore, we would like to acknowledge EUMETSAT for making Dust RGBs

available for download. The in-situ data used are part of the Norwegian national monitoring program (Aas et al., 2020) funded by the Norwegian Environment Agency, except the tracer analysis at Zeppelin which is funded by the Ministry of Climate and Environment, while tracer analysis at Birkenes is funded by NILU. The research leading to these results has benefited from Aerosols, Clouds, and Trace gases Research InfraStructure (ACTRIS), funding from the European Union Seventh Framework Programme (FP7/2007–2013) under ACTRIS-2 and the grant agreement no. 262254, and the COST Action CA16109,

Chemical On-Line cOmpoSition and Source Apportionment of fine aerosol (COLOSSAL).

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



**Appendix A1: Sample analysis**

OC/EC was obtained by Thermal-optical analysis (TOA), using the Lab OC-EC Aerosol Analyser (Sunset Laboratory Inc) and according to the EUSAAR-2 temperature program (Cavalli et al., 2010). The samples content of $CO_3^{2-}$- carbon was determined as for OC and EC but after subjecting a punch of the filter sample to thermal-oxidative pre-treatment (Jankowski et al., 2008; Evangeliou et al., 2016). The samples OC and EC content were corrected with respect to the $CO_3^{2-}$- carbon.

Organic tracers (levoglucosan, mannosan, galactosan, arabitol, mannitol, glucose, trehalose, 2-methylerythritol and 2-methylthreitol) were analyzed using Waters Acquity ultra-performance liquid chromatography (UPLC) in combination with Waters Premier XE high-resolution time-of-flight mass spectrometry (HR-TOF-MS) operated in the negative electrospray ionization (ESI) mode. The analytical methodology is based on that of Dye and Yttri (2005) but deviates by choice of column (two 2.1_150mm HSS T3, 1.8 μm, Waters Inc.). All species were identified by retention time and mass spectra of authentic standards and isotope-labelled standards of levoglucosan, galactosan, mannitol, arabitol, trehalose and glucose were used as recovery standard (see Yttri et al., 2021).

Al, Fe, Mn and Ti were analyzed by ICP-MS (Inductively Coupled Plasma/Mass Spectrometry) (Agilent 7700x). Prior to analysis, aerosol filter samples were extracted (diluted supra pure $HNO_3$), digested (75 min; max temp of 250 ˚C for 15 min) using an UltraClave microwave system (Milestone, Italy), and diluted to 10 ml (ion exchanged $H_2O$). External calibration was applied, and calibration standards made of $HNO_3$ (supra pure) (10% v/v) to adapt to the sample matrix. Indium was used as recovery standard and applied to all samples, standards, blank filters, and reference materials.

Prior to ion chromatography analysis, filter samples were soaked in Milli-Q water (10 ml) and subjected to ultrasonic agitation (30 min). Extracts were analyzed with respect to $Ca^{2+}$, $K^+$, $Mg^{2+}$, $Na^+$, and $NH_4^+$ on a Dionex Integrieon ion chromatograph, using a Dionex cation exchange CS16 column (3 mm x 250 mm), and a conductivity detector. Samples were eluted using methane sulphonic acid (34 mM) at a flow rate of 0.5 ml min$^{-1}$. $Cl^-$, $NO_3^-$, and $SO_4^{2-}$ were analyzed on a Dionex Integrion ion chromatograph, using a Dionex anion exchange AS9-SC column (4 mm x 250 mm), and a conductivity detector. Samples were eluted using carbonate ($K_2CO_3$, 2.0 mM; $HCO_3^-$, 0.75mM) at a flow rate of 2 ml min$^{-1}$.