# Peer review of "What caused a record high $PM_{10}$ episode in northern Europe in October 2020?"

_Atmospheric Chemistry and Physics, 2021_

## Author Response (AR1)

Reviewer 1

The paper by Groot Zwaaftink et al. investigates a recent (October 2020) record-breaking $PM_{10}$ episode in northern Europe with the aid of surface measurements, satellite observations, and modelling approaches. This is a very interesting and comprehensive in-depth analysis exploring the sources and mechanisms that resulted in several exceedances of the EU $PM_{10}$ limits over northern Europe. The paper is overall well written, and I really liked the sequence of analysis steps. As apart from local sources, long range transport is also responsible for PM standard exceedances, such studies unravel limitations of modeling and forecast systems, contributing to the direction of improving air quality management. Thus, I support publication of the paper after the following comments are considered by the authors.

*Thank you for your constructive review which helps us improve our manuscript.*

Comments:

1. Please use a), b), .. indexing for all Figures and modify accordingly the manuscript. This will be very helpful for the reader.

   *We have adapted the figures and manuscript accordingly.*

2. P3 Figure 1: Since there is no distinct color separation for values > 50 µg/m3 (left) and >25 µg/m3 (right), the reader cannot distinguish the sites where the air quality limits are exceeded. I suggest to use less color levels in a way that colors from red and on are referring to air quality limit exceedances. Another option would be to plot station exceedances with another marker (instead of circle you can use + for example). Moreover, I suggest to provide 2 and 3 of October 2020 as separate Figures (4 in total) in order to have sense the visualization of exceedances.

   *We have reduced the number of colours and split the data to separate figures for 2 and 3 October 2020 to better visualize exceedances of air-quality limits.*

3. P17 Figure 5: I understand the choice of the dates. Yet, for a more comprehensive view of the event and to better unravel the pathways of BC and dust transport, apart from start (27/9) and end (02/10) of the episode the intermediate dates are also essential. I suggest to provide (maybe as Supplement) the respective maps from 27/9 to 02 or 03/10 with a 6-hour (or 12-hour) interval, as a Figure or animation. Also, I suggest using another color for coastlines, as it is difficult to distinguish from contour lines and color-shaded.

   *Animations of FLEXPART simulated dust and BC transport from 27 September to 8 October are added as supplements. We changed the colours of the coastlines to white.*

4. As the state-of-the-art CAMS-global (IFS) forecast system provides aerosol species for both dust (3 bins) and black carbon, it would be interesting to investigate if it has an added value to the existing analysis.

   *Quantitative results considering dust concentrations from CAMS are already included. BC from biomass burning and BC from fossil fuels are not included in the*

*CAMS forecast system, only PM10 from wildfires or total elementary carbon. A comparison to those will unnecessarily complicate the analysis.*

*Maps of the dust and BC plumes based on CAMS, that are rather illustrative to explain the episode, do not add much to the current analysis based on FLEXPART simulations, in our opinion.*

Minor Comments:

*We here omit minor comments where we changed the manuscript following the reviewer's suggestions.*

- P1, L11-12: This sentence needs reconstruction. I guess on-line and off-line refers to transport model simulations and not to observations. Please make this clear.

Changed to: *"We analysed this severe pollution episode caused by long-range atmospheric transport based on surface and remote sensing observations and transport model simulations to understand its causes."*

- P9, L251: The CAMS regional forecast product is for 4 days (96h) in advance.

*We made an error here and thank the reviewer for identifying this. The forecast does indeed go out to 96 hours. However, since we only use the first 24 hours of each forecast from each day this does not impact the results we show.*

**Reviewer 2**

This is an interesting case study on the origin of an episode of high PM in Norway (and Northern countries in general). The authors do a good job in describing the introduction, methods, and analysis. In my minor comments, I have some requests for clarifications, which should be easy to solve. As a major comment, I would question however, what have we really learned on LRT of dust and BC, that we didn't know before? For instance, it would be good to know whether this unique event (at least on a 4-10 year timescale), was and will remain exceptional, are there signs of intensification of emissions, change in transport patterns that could seriously jeopardize air quality in Norway (and other surrounding countries). While I realize that such analysis is beyond scope of the current paper, I do think that the paper could be strengthened, by outlining what further analysis should be done, to answer such questions. The paper also identified rather big differences between CAMS regional model efforts, and FLEXPART. What have we learned from these differences, and what would be steps to improve models?
I therefore complement the authors for the work in this publication, but also invite them to discuss above, to improve the relevance for a wider audience.

*Thank you for your constructive comments. Indeed the question how unique this event was and will remain is very interesting. This could be investigated based on analysis of transport patterns over past and future years and trend analysis of emissions of mineral dust and wildfires in earth system models. Such analysis would go beyond the scope of this paper but we now included a short discussion in the manuscript:*

*"Further research is needed to assess whether there are reasons to assume that this kind of episodes may occur more frequently in future. Emissions from wildfires and mineral dust sources are sensitive to changes in climate, land use and human activities and different scales should be considered. Also, changes in atmospheric transport patterns could affect the occurrence of LRT-episodes in Europe. Future studies should thus include earth system model simulations to give us a better understanding in the occurrence of such episodes in future, although especially the complexity due to human activities will make it difficult to draw conclusions." (p34)*

*We understand that one would like to see steps to improve the models in the CAMS regional model efforts, but this was not an aim of the current study. This would require careful analysis of emission amounts and transport parameterizations in each of the models, which we cannot provide here. However, we added a comment on possible problems or differences between the models: "Furthermore, likely causes of differences between the models are for instance included dust sources, emission and scavenging parameterizations." (p24)*

Detailed comments

l. 9 awkward sentence. Suggest: recorded weekly values exceeded historical weekly maxima for at least 4 up to 10 years. Why is a comparison made on weekly timescales and not on daily?

> *Changed accordingly. We include weekly values to be consistent with the weekly samples taken at background sites.*

> l. 11 what does the on-line /off-line refer to? Models? Please clarify

> *It refers to different measurements, we removed both terms for simplification.*

l. 17 expand the 'can not exclude other contributing sources'. Why can not excluded, and what could the 'other sources be.

> *Changed to: "The biomass burning fraction can be attributed to forest fires in Ukraine and southern Russia, but we cannot exclude other sources contributing, like fires elsewhere, because the model underestimates observed concentrations."*

L 103. I did not find additional information on this assumption in Appendix 1. Can a proper analysis on solubility be given in the appendix, and also why the specific associated ion rates choice.

l. 105 Sure it is an approximation, but is it a good approximation? Indeed reactions do take place, and it would be good if in further analysis later in this paper the validity of these assumptions will be tested- beyond the equation 2 and 3.

> *L 103: The reference made to Appendix 1 is for a description of the ion chromatography method used, not for the $CO_3^{2-}$ speciation. We have clarified this in the revised document:*

> *New:*

*(See Appendix 1 for a description of the ion chromatography method)*

*L103 and L105: We choose to merge our answer to ref.2's comment on L103 regarding solubility and on L105 regarding reactions taking place in the atmosphere as these both are related to the estimate of the samples mineral dust content.*

*Solubility is a function of pH, which is an unknown variable here. Ion equivalent calculations suggest that all $SO_4^{2-}$ and $NO_3^-$ were associated with $NH_4^+$, thus $H^+$ is not available, indicating a neutral condition. This also indicates that reactions between $CO_3^{2-}$ and $HNO_3^-$ and/or $H_2SO_4$ are not profound. If formation of nitrates and sulphates take place, resulting from reactions between $CO_3^{2-}$ of mineral dust origin and $HNO_3/H_2SO_4$, it is also a matter of discussion whether these species should be apportioned to the mineral dust fraction.*

*Regarding the solubility of $CaCO_3$, $MgCO_3$ and $K_2CO_3$, $K_2CO_3$ is considered soluble in $H_2O$ at 20 ˚C (Table 1) and thus should be considered part of the water-soluble fraction determined by ion chromatography. We have changed the original sentence to account for this:*

*Original:*

*We also made assumptions regarding the speciation of $CO_3^{2-}$, assuming 70% was present as $CaCO_3$, 20% as $MgCO_3$ and 10% as $K_2CO_3$. $Ca^{2+}$, $Mg^{2+}$, and $K^+$ assumed to be associated with $CO_3^{2-}$ were not considered part of the water-soluble fraction determined by ion chromatography (Appendix 1).*

*New:*

*We also made assumptions regarding the speciation of $CO_3^{2-}$, assuming 70% was present as $CaCO_3$, 20% as $MgCO_3$ and 10% as $K_2CO_3$. $Ca^{2+}$ and $Mg^{2+}$ assumed to be associated with $CO_3^{2-}$ were not considered part of the water-soluble fraction determined by ion chromatography (Appendix 1).*

Table 1: Solubility of carbonates, nitrates, oxides, and sulphates in $H_2O$ at 20 ˚C.

| | $CO_3^{2-}$ | $NO_3^-$ | $SO_4^{2-}$ |
|---|---|---|---|
| $K^+$ | HS (111 g ml$^{-1}$) | HS (31.6 g ml$^{-1}$) | HS (11.1 g ml$^{-1}$) |
| $Mg^{2+}$ | IS (0.039 g ml$^{-1}$) | HS (69.5 g ml$^{-1}$) | HS (35.1 g ml$^{-1}$) |
| $Ca^{2+}$ | IS (0.00062 g ml$^{-1}$) | HS (129 g ml$^{-1}$) | PS (0.255 g ml$^{-1}$) |

IS = Insoluble (< 0.01 g pr. 100 g $H_2O$); PS = partly soluble (between 0.01 g and 1.0 g pr. 100 g $H_2O$);

HS = Highly soluble (> 1.0 g pr. 100 g $H_2O$); Rx = reacts with $H_2O$

*Further, we accounted for the solubility of $K_2CO_3$ in $H_2O$ at 20 ˚C in our calculations examining the impact of $CO_3^{2-}$ speciation (70% $CaCO_3$:20% $MgCO_3$:10% $K_2CO_3$) on the mineral dust (MD) estimate and when examining the impact that reactions between $CO_3^{2-}$ and $HNO_3$ and $H_2SO_4$ might have on the MD estimate.*

*Regarding the impact of $CO_3$ speciation on MD estimate (L103):*

*The $CO_3^{2-}$ – fraction (including the associated $Ca^{2+}$, $Mg^{2+}$, $K^+$) makes a modest (6.6 – 12%) increase to the estimated lower MD concentration (consisting of $SiO_2$, $Al_2O_3$, $Fe_2O_3$, $TiO_2$, and $MnO$), assuming 70% of $CO_3$ is $CaCO_3$, 20% is $MgCO_3$ and 10% is $K_2CO_3$.*

*By assuming that all $CO_3^{2-}$ was present as $MgCO_3$ (providing the lowest estimate of the $CO_3^{2-}$ fraction), the $CO_3^{2-}$ – fraction would make a 5.5 – 10% contribution to the lower MD estimate. $MgCO_3$ is insoluble at neutral pH, thus we cannot use the observed concentration of water-soluble $Mg^{2+}$ to validate if this is theoretically possible. Assuming all $CO_3^{2-}$ was present as $K_2CO_3$ would give the highest estimate of the*

*$CO_3^{2-}$ fraction. $K_2CO_3$ is water soluble and only at Zeppelin was the observed concentration of water-soluble $K^+$ high enough to balance the observed $CO_3^{2-}$. We find that the $CO_3^{2-}$ fraction would make a 10% contribution to the lower MD estimate at Zeppelin if all $CO_3^{2-}$ was present as $K_2CO_3$, 7.5% assuming the 70%:20%:10% split, and 6.3% if all was present as $MgCO_3$. Hence, the speciation of the $CO_3$-fraction has a minor impact on the estimated MD concentration.*

*We have included the following sentence in the revised paper to highlight this:*

*The exact $CO_3^{2-}$ speciation is however of minor importance, as the $CO_3^{2-}$ - fraction increases the lower mineral dust estimate (eq.2) by only 6.3% (assuming all is $MgCO_3$) to 10% (assuming all is $K_2CO_3$), using data for Zeppelin as an example.*

*Regarding the potential impact of reactions between $CO_3^{2-}$ and $HNO_3/H_2SO_4$ on the estimated MD concentration (L105):*

*Reactions between $[Ca^{2+}, Mg^{2+}, K^+]CO_3$ and $HNO_3/H_2SO_4$ forms $CO_2$, $H_2O$, $[Ca^{2+}, Mg^{2+}, 2K^+](NO_3)_2$ and $[Ca^{2+}, Mg^{2+}, 2K^+](SO_4)$, e.g.:*

$$CaCO_3 \text{ (s)} + HNO_3 \text{ (g)} = Ca(NO_3)_2 \text{ (aq)} + H_2O \text{ (l)} + CO_2 \text{ (g)} \qquad \text{(eq. 1)}$$

$$CaCO_3 \text{ (s)} + H_2SO_4 \text{ (g)} = CaSO_4^{2-} \text{ (aq)} + H_2O \text{ (l)} + CO_2 \text{ (g)} \qquad \text{(eq. 2)}$$

*$CaNO_3$, $MgNO_3$ and $K_2NO_3$ are all soluble in water at 20 ˚C, as is $MgSO_4$ and $K_2SO_4$. By assuming that water soluble $Ca^{2+}$, $Mg^{2+}$ and $K^+$ not attributed to sea salt aerosol (or to $K_2CO_3$ in the case of $K^+$) originates from reactions between $[Ca^{2+}, Mg^{2+}, K^+]CO_3$ and $HNO_3/H_2SO_4$, limited by observed concentrations of $NO_3^-$ and $SO_4^{2-}$, and that excess water-soluble $Ca^{2+}$, $Mg^{2+}$ and $K^+$ (i.e. not associated with $NO_3^-$ and $SO_4^{2-}$) is present as oxides, we calculate a maximum estimate (column entitled "Incl. rx between $CO_3^{2-}$ and $H_2SO_4/HNO_3$") of the MD concentration that is somewhat higher than the "Upper estimate" presently shown in Zwaaftink et al. (in rev.) where all water soluble $Ca^{2+}$, $Mg^{2+}$ and $K^+$ not attributed to sea salt aerosol or $K_2CO_3$ are related to oxides. There are negligible differences in the calculated concentrations regarding whether $Ca^{2+}$, $Mg^{2+}$ and $K^+$ are associated with $NO_3^-$ or $SO_4^{2-}$ first, although the speciation will differ.*

Table 2: Potential constituents of mineral dust.

| | "Lower estimate" in Zwaaftink et al | | | | "Upper estimate" in Zwaaftink et al | Incl. rx between $CO_3^{2-}$ and $H_2SO_4/HNO_3$ |
|---|---|---|---|---|---|---|
| | $Fe_2O_3$; $Al_2O_3$; $TiO_2$; MnO | CaO; MgO; $K_2O$ | $CaCO_3$; $MgCO_3$; $K_2CO_3$; | $Ca(NO_3)_2$; $Mg(NO_3)_2$; $KNO_3$; $MgSO_4$; $K_2SO_4$ | Me-Oxides; Carbonates; Oxides | |
| Birkenes Observatory | 6.2 | 0.89 | 0.41 | 2.4 | 7.6 | 9.1 |
| Hurdal | 5.6 | 0.94 | 0.66 | 1.8 | 7.2 | 8.1 |
| Kårvatn | 7.6 | 1.2[1] | 0.93 | 2.8[1] | 9.8 | 11.4 |
| Zeppelin Observatory | 1.9 | 0.50 | 0.15 | 0.96 | 2.6 | 3.0 |

1) Data constructed based on relative contribution of these fractions at Birkenes and Hurdal.

*To meet the request from the referee asking how good our approximation is we have included the following text in the revised manuscript:*

*Original text:*

*...not part of sea salt aerosol) were assumed to be present as oxides (CaO, MgO, and K₂O). We consider this an approximation, as chemical reactions likely take place during atmospheric transport. Both $NO_3^-$ and $SO_4^{2-}$ were enhanced during the episode and could indicate the presence of $Ca(NO_3)_2$ and $CaSO_4$ formed from reactions between $CaCO_3$ and acids such as $HNO_3$ or $H_2SO_4$ (Laskin et al., 2004).*

*New text:*

*...sea salt aerosol or $K_2CO_3$) were assumed to be present as oxides (CaO, MgO, and K₂O), which together with the carbonates and metal oxides provided the upper mineral dust estimate (eq.3). We consider this an approximation, as chemical reactions likely take place during atmospheric transport. Both $NO_3^-$ and $SO_4^{2-}$ were enhanced during the episode and could indicate the presence of e.g. $Mg(NO_3)_2$ and $MgSO_4$ formed from reactions between $MgCO_3$ and acids such as $HNO_3$ or $H_2SO_4$ (Laskin et al., 2004). Accounting for such reactions would increase the upper mineral dust estimate (eq.3) by 12 – 21%. However, it is not apparent that nitrates and sulphates formed this way should be apportioned to the mineral dust fraction. Finally, ion equivalent calculations suggest that all $SO_4^{2-}$ and $NO_3^-$ were associated with $NH_4^+$, thus we do not include these potential reactions in the upper estimate of mineral dust.*

l. 115 Clarify if consequently the equations imply an upper and lower limit for biomass burning aerosol, as per mass balance. I think this is done in equation 8.

*It is not entirely clear to us what the referee asks for with respect to line 115. Still, section 2.1.2 explains the rationale for the quantitative estimate of the mineral dust fraction, whereas the corresponding explanation for the biomass burning (BB) aerosol fraction is in section 2.1.3. The lower and the upper estimate of the BB fraction is provided in eq.7 and eq.8. Line 109 – 115 explains that the carbonates and the oxides can be associated with both mineral dust and BB and it is appropriate to mention this in section 2.1.2 as this one comes first. Indeed, the upper estimate of mineral dust contains all oxides (not the metal-oxides) and carbonates, thus it cannot at the same time be apportioned to the BB fraction. Likewise, the upper estimate of the BB includes all oxides and carbonates and thus cannot at the same time be apportioned to the mineral dust fraction. To make this mutual dependency on the upper estimate of the two fractions clearer, we have included the following sentence:*

*New text:*

*With $CO_3^{2-}$ and oxides apportioned to mineral dust (eq.3) an upper estimate of BB (eq.8) would not be possible and vice versa.*

l. 130 as emission ratios vary widely, did the authors consider making an sensitivity analysis?

*In the text we state that "Emission ratios of levoglucosan from wildfires are likely to vary widely reflecting combustion conditions and vegetation, and source region." By applying emission ratios derived from ambient sampling of wildfires emissions in Eastern Europe, including Ukraine we do account for the variability in all the variables mentioned (combustion conditions, vegetation, and source region) in the best possible way. Including measurements from other regions, exhibiting different vegetation, would in our view be a suboptimal approach, and even wrong, introducing data that is not representative for the actual region and thus including unnecessary uncertainty to the calculations.*

133 what is meant with atmospheric depletion? Oxidation?

*We think that atmospheric "degradation of levoglucosan" would make the sentence clearer to the reader.*

*New text:*

*"Calculated concentrations of $OC_{BB}$ and $EC_{BB}$ should be considered semiquantitative given the uncertainty of the emission ratios and the potential atmospheric degradation of levoglucosan."*

l. 139 what is the meaning of (TC/levoglucsan)bb I guess this is the emission ratio, but please specify.

*Yes, it is an emission factor. We have added the following sentence to clarify this:*

*New text:*

*For eq.4, eq.5 and eq.10, notations in parentheses are emissions ratios.*

l. 140 same for (OC/TC)bb

*See reply to l.139*

l. 149/150 if this an important assumption, sensitivity analysis is needed.

*The use of a conversion factor is standard procedure in numerous studies where the carbon mass concentration of OC is to be converted to mass concentration of OM. This factor is a linear factor, for which a sensitivity analysis appears superfluous.*

l. 164 I expect that ECff is the difference of two big numbers, and therefore highly uncertain- at least in certain periods. Can the authors analyse the associated uncertainty?

*$EC_{FF}$ is not the difference between two big numbers. $EC_{FF}$ results from eq.9*

$[EC_{FF}] = [EC] - [EC_{BB}]$ (eq.9)

*and the mean $EC_{BB}/EC$ ratio was 0.5, 0.5, 0.53, and 0.65 at the four sites, thus the concentrations of $EC_{bb}$ and $EC_{ff}$ are comparable. The uncertainty of $EC_{FF}$ is however connected to the uncertainty related to $EC_{BB}$, which is discussed on lines 473 – 475 in the original draft when the $EC_{BB}/EC_{FF}$ split is compared to the $eBC_{BB}/eBC_{FF}$ split at the Birkenes Observatory:*

*"The $eBC_{BB}/eBC_{FF}$ split is thus comparable to the levoglucosan approach, which apportioned equally large shares to $EC_{BB}$ and $EC_{FF}$ for 30 September-7 October but note that the range (50±20%) of the levoglucosan approach is very wide."*

*This paragraph thus shows that for the Birkenes Observatory, the $EC_{BB}$ fraction is 50±20%. Based on the relationship between $EC_{BB}$ and $EC_{FF}$ expressed in eq.9, a similar estimate is valid for $EC_{FF}$.*

l. 187 something seems missing "1OBJ"

*Corrected*

l. 215 Give some further information on what aspects can be retrieved by running Flexpart in forward and backward mode, and why a different use is made for BC and mineral dust.

*Backward simulations can be computationally efficient to distinguish source regions for particular observations. A difference between BC and mineral dust is made here because the mineral dust simulations include 10 size bins, which can be combined in a single forward run, but would need separate backward runs. We comment on this in the manuscript: "We chose this setup rather than backwards simulations (like we did for BC) because backwards simulations for mineral dust would have required separate simulations for each size bin and station and this approach was thus more efficient." (p10)*

229 why was 1x1 degree resolution used, which is later refined in 0.5 degree output. Simulations should be available on even finer resolution.

*FLEXPART is a Lagrangian model and the output grid can be defined differently from the meteorological input. The chosen resolutions were a compromise between computational efficiency and accuracy.*

249-259 To what extent would these regional models be able to pick up dust emissions from central Asia, as the domain only partly covers the Central Asian desert region, and I guess the link to boundary conditions from the global model is not so straightforward.

*Each of the regional models within the ensemble uses the boundary conditions from the global IFS model, and this includes the IFS dust concentrations represented in three size bins. By this method, dust from, e.g., central Asia, can be represented in the regional domain. As far as we know, each regional modelling team applies an appropriate scaling methodology (known error in MOCAGE notwithstanding) between the IFS bins and the regional model bins. We do acknowledge that this is a potential source of uncertainty now in an additional comment (see response to comment on line 410 below).*

163 Define RGB (definition come too late).

*Changed accordingly.*

270 clarify this is top of the atmosphere radiance as observed by OLCI

*This has been clarified.*

273-280 clarify where the values used for the simulations were coming from.

*This has been clarified.*

293 days means exceedance days? The statement is somewhat confusing to give a number of exceedance days multiplied by # of stations.

*Yes, we added 'exceedance' to clarify.*

295 to make sure I get this right: the number of 2 exceedance in November, makes up almost 20 % of what is usually (208;2019) observed?

*Yes, over these 2 days the number of exceedances added up to 18% of the total number of total exceedance days in previous years. Changed to:"In comparison to years 2018 and 2019, for a selection of 30 sites with measurements in those years, this means that the number of exceedance days during the episode corresponded to 18% of the average total exceedance days for 2018 and 2019. "*

306 already? What is meant?

*Removed.*

310 explicitly mentionwheter a closure between 73 and 93 % is satisfactory or not, and in line/or not with other pubished results.

*A minor correction changed the mass closure range from 73 – 93% to a corrected to 74 – 92%. In accordance with the request from the referee we have added the following sentence to the revised version of the draft:*

*"The range of mass closure obtained for the sites is comparable to previous studies (e.g. Putaud et al., 2010; Yttri et al., 2021; Aas et al., 2021) but should be considered a conservative estimate, using the lower estimate of the mineral dust fraction."*

*We have included the following references to the reference list:*

*Putaud, J.-P., Van Dingenen, R., Alastuey, A., Bauer, H., Birmili, W., Cyrys, J., Flentje, H., Fuzzi, S., Gehrig, R., Hansson, H. C., Harrison, R. M., Herrmann, H., Hitzenberger, R., Hüglin, C., Jones, A. M., Kasper-Giebl, A., Kiss, G., Kousa, A., Kuhlbusch, T. A. J., Löschau, G., Maenhaut, W., Molnar, A., Moreno, T., Pekkanen, J., Perrino, C., Pitz, M., Puxbaum, H., Querol, X., Rodriguez, S., Salma, I., Schwarz, J., Smolik, J., Schneider, J., Spindler, G., ten Brink, H., Tursic, J., Viana, M., Wiedensohler, A., and Raes, F.: A European aerosol phenomenology – 3: Physical and chemical characteristics of particulate matter from 60 rural, urban, and kerbside sites across Europe, Atmos. Environ., 44, 1308–1320, 2010.*

*Aas, W., Eckhardt, S., Fiebig, M., Platt, S.M., Solberg, S., Yttri, K. E., and Zwaaftink Groot, C.: Monitoring of long-range transported air pollutants in Norway, annual report 2020, Miljødirektoratet rapport, NILU, Kjeller, Norway, M-2072/2021 NILU OR 13/2021, 2021.*

324 the long-term mean 2016-2019 refers to the annual average of daily/weekly over these years?

*This refers to the mean values for September-November in years 2014 to 2019. We added "seasonal" to emphasize that these are not annual averages.*

346 contribution of what?

*Changed to "mass contribution"*

365 simulations refers to the emission module?

*Yes, changed to FLEXDUST simulations.*

391 it would be useful to include at some lines some labels for flexpart in the plots- it is quite

tedious to go back and forward to the legenda. Also I am wondering to what extent the coincidence of patterns is really a sign of agreement, the text could be more expanded to explain if also radiative closure is obtained.

*The comparison shown in Figure 6 is a qualitative comparison, showing the agreement/disagreement of aerosol layer, i.e., dust and BC, in the troposphere. From CALIOP, we show the aerosol extinction at 532 nm, while for FLEXPART mineral dust and BC concentrations are shown. It was not intended, nor is it suitable for a comparison in terms of radiative properties. Our focus in on the transport/location of the respective layer, not their optical/radiative properties. A legend was added to the figure.*

410 if there is a clear error identified the result should probably not be used (MOCAGE). It remains unclear why other CAMS models are so much lower than Flexpart. In general wet deposition is an important cause for discrepencies. If I understand well the resolution of CAMS models and Flexpart is the same/similar around 1x1 degree?

*The regional ensemble forecast is a product provided directly by CAMS and includes all of the regional models as it is and without any input or modification by the authors. We show the median of all ensemble members because this itself is the state-of-the art forecast product, which is being widely used in local-scale air quality modelling applications, and for air quality assessments for policymakers. Due to this, we think there is a valid scientific motivation for testing its performance. If this raises a problem with specific aspects of the data product, as we found in this case, then reporting this issue should be of interest to potential interest to its users. We identify here specific problems we are aware of having contacted the relevant modelling team, but do not aim for this to be an extensive model comparison as it goes beyond the scope of the article. (Instead we have added the following text to the manuscript: "Furthermore, likely causes of differences between the models are for instance included dust sources, emission, treatment of boundary conditions and scavenging parameterizations.").*

425 looking at the Birkeness data, it seems that also flexpart is grossly underestimating dust (assuming that in the period 2-4 October, dust is the dominant PM10 component). Interestingly there is a peak in the models that is not visible in the observations- end of September, although levels seem to be comparable. Not clear how dust concentrations in Birkeness lower panel (around 7 ug/m3 are related to the much higher PM10 levels). Looks inconsistent?

*If dust were the dominant PM10 component in this period that would be correct. But there was a strong contribution from BC as well with a different time signal, as we know from the chemical analysis and figure 8. This complicates the comparison of PM10 and modelled dust time series and we cannot verify whether the modelled peak dust concentrations end of September actually occurred. However, the modelled dust values do not exceed the observed PM10 values and the dust peak is thus likely to be simply obscured by other signals and species in the total PM10 observations.*
*Although peak PM10 concentrations are much stronger, the average of PM10 concentrations shown in the top panel in the period 30 September to 7 October (covered by the sample in the bottom panel) was only 20.2 ug/m3. A mean dust concentration of around 7*

*ug/m3 is consistent with this value, given the additional large contribution from BC to PM10 in this period.*

483 how much underestimated?

*Roughly a factor 10 (added to manuscript).*

521 please elucidate a bit better what is displayed in Figure 10a (with unit per second).

*We added: "The SRR indicates how sensitive the concentrations at the receptor are to emissions in different source regions."*